# Recent Advances in Genomics-Based Approaches for the Development of Intracellular Bacterial Pathogen Vaccines

**DOI:** 10.3390/pharmaceutics15010152

**Published:** 2022-12-31

**Authors:** Muhammad Ajmal Khan, Aftab Amin, Awais Farid, Amin Ullah, Abdul Waris, Khyber Shinwari, Yaseen Hussain, Khalaf F. Alsharif, Khalid J. Alzahrani, Haroon Khan

**Affiliations:** 1Division of Life Science, Center for Cancer Research, and State Key Lab of Molecular Neuroscience, Hong Kong University of Science and Technology, Hong Kong, China; 2Division of Environment and Sustainability, Hong Kong University of Science and Technology, Hong Kong, China; 3Molecular Virology Laboratory, Department of Microbiology and Biotechnology, Abasyn University, Peshawar 25000, Pakistan; 4Department of Biomedical Sciences, City University of Hong Kong, Hong Kong, China; 5Institute of Chemical Engineering, Department Immuno-Chemistry, Ural Federal University, Yekaterinbiurg 620002, Russia; 6Department of Pharmacy, Abdul Wali Khan University Mardan, Mardan 23200, Pakistan; 7Department of Clinical Laboratory, College of Applied Medical Science, Taif University, P.O. Box 11099, Taif 21944, Saudi Arabia

**Keywords:** vaccine development, genomes, intracellular bacteria, transcriptomics, proteomics

## Abstract

Infectious diseases continue to be a leading cause of morbidity and mortality worldwide. The majority of infectious diseases are caused by intracellular pathogenic bacteria (IPB). Historically, conventional vaccination drives have helped control the pathogenesis of intracellular bacteria and the emergence of antimicrobial resistance, saving millions of lives. However, in light of various limitations, many diseases that involve IPB still do not have adequate vaccines. In response to increasing demand for novel vaccine development strategies, a new area of vaccine research emerged following the advent of genomics technology, which changed the paradigm of vaccine development by utilizing the complete genomic data of microorganisms against them. It became possible to identify genes related to disease virulence, genetic patterns linked to disease virulence, as well as the genetic components that supported immunity and favorable vaccine responses. Complete genomic databases, and advancements in transcriptomics, metabolomics, structural genomics, proteomics, immunomics, pan-genomics, synthetic genomics, and population biology have allowed researchers to identify potential vaccine candidates and predict their effects in patients. New vaccines have been created against diseases for which previously there were no vaccines available, and existing vaccines have been improved. This review highlights the key issues and explores the evolution of vaccines. The increasing volume of IPB genomic data, and their application in novel genome-based techniques for vaccine development, were also examined, along with their characteristics, and the opportunities and obstacles involved. Critically, the application of genomics technology has helped researchers rapidly select and evaluate candidate antigens. Novel vaccines capable of addressing the limitations associated with conventional vaccines have been developed and pressing healthcare issues are being addressed.

## 1. Introduction

The majority of infectious diseases that affect humans are caused by bacteria [1]. Despite recent advances in antimicrobial medicine, bacterial infections are still a major cause of mortality worldwide. On the basis of their infective lifestyle, pathogenic bacteria are classified into three major groups: extracellular, facultative intracellular, and obligatory intracellular. Bacteria that live outside the host are known as extracellular or free-living bacteria, while those that have both intracellular and extracellular phases (capable of replicating inside and outside a host cell) are known as facultative intracellular bacteria [1]. Obligatory intracellular pathogenic bacteria are distinct from the preceding two bacteria in that they have restricted characteristics that necessitate a host for reproduction [1]. A better grasp of the complicated interplay between hosts and pathogens will help create new preventive and therapeutic treatments that can be applied to resolve infection. Accurate classifications of infectious agents have been developed that take the infective lifestyle of various pathogens and their specificities of hosts into consideration [2]. For the purpose of organizing fruitful future research, this knowledge is crucial.

Many bacteria, such as *Escherichia coli* (*E. coli*), *Haemophilus influenzae* (*H. influenzae*), *Mycoplasma* spp., *Pseudomonas aeruginosa* (*P. aeruginosa*), *Streptococcus pyogenes* (*S. pyogenes*), *Vibrio cholerae* (*V. cholerae*), etc., are extracellular, possessing the ability to reproduce in extracellular space. Extracellular bacteria are normally eliminated from the body by macrophages (M), neutrophils, and dendritic cells (DCs) through phagocytosis. However, those that survive may potentially be hazardous, as they release toxins and trigger excessive inflammatory responses, damaging and destroying host cells and tissues. Comparatively, intracellular bacteria have developed adaptations that allow them to avoid lysosome-mediated degradation in host cells, allowing them to replicate. The body has evolved distinct immune responses, such as T cell activation, particularly CD8+ T cells, to resolve such infections. Facultative intracellular bacteria can survive both inside and outside of host cells. They include *Legionella pneumophila* (*L. pneumophila*), *Salmonella* spp., *Mycobacterium tuberculosis* (*M. tuberculosis*), *Neisseria meningitidis* (*N. meningitidis*), *Shigella dysenteriae* (*S. dysenteriae*), *Francisella tularens* (*F. tularens*), *Bordetella pertussis* (*B. pertussis*), and *Bacillus anthracis* (*B. anthracis*). On the other hand, obligate intracellular bacteria are wholly dependent on the host cell machinery for their existence and division. *Chlamydia* spp., *Anaplasma* spp., *Ehrlichia* spp., *Coxiella* spp., and all rickettsial and Orientia species are examples of obligate intracellular bacteria. These bacteria cause fatal diseases in humans. The incubation period for different bacteria varies. Some bacteria such as *S*. *flexneri*, *S. typhimurium*, and *S. enterica* have a short 12 h incubation, while others such as *M. leprae* have longer incubation periods that may last 4–5 years (Table 1). It is important to note that alternative pathogen classification methods, based on criteria such as molecular typing, 16S rRNA, and phylogenetics also exist in the literature. Conventional classifications are primarily based on the infective lifestyles of pathogens in the host; however, there are many inconsistencies in this approach. For example, some extracellular bacteria are known to also have an intracellular in vivo phase. The ability of such pathogens to grow in artificial cell-free media has often been exaggerated [2]. In this context, the term “facultative” has not been adequately explained, as the ability of a pathogen to grow intracellularly during infectious processes has been termed facultative, while in reality, the pathogen characteristics are that of obligate bacteria, and therefore essential. In contrast the term “obligate” is based on the inability to survive in artificial conditions [2]. This particular inconsistency in bacterial classification has since been resolved by researchers who examined the role of peptidoglycan recognition proteins (PGRPs) in *Drosophila* [3,4]. PGRPs are pattern recognition molecules that discern unique peptidoglycans in bacterial cell walls. PGRP-LE and PGRP-LC in particular, are involved in the identification of extracellular and intracellular bacteria that have diaminopimelic acid (DAP) type peptidoglycans. They trigger a variety of innate immune responses, including the generation of antimicrobial peptides, melanization, and autophagy.

Vaccines and antibiotics have significantly improved global health. Vaccines prevent fatal infections, while antibiotics have been pivotal in the treatment of life-threatening and often fatal infectious diseases caused by intracellular bacteria. The development of vaccines against intracellular microorganisms faces numerous difficulties. The genetic diversity found in bacterial strains poses one of the most profound obstacles. The emergence of antibiotic resistance is another challenge that further increases the genetic diversity of bacteria. Furthermore, as the capsular polysaccharide of bacteria has weak immunogenicity and high similarity to fetal neural tissues, the development of vaccines against bacteria such as *N. meningitidis* (serogroup B vaccine) is more challenging. The rise in intracellular bacteria that are resistant to inactivated whole cell vaccines also presents an additional layer of difficulty in vaccine development. Therefore, to overcome these difficulties and challenges, novel approaches are required for the development of anti-intracellular bacteria vaccines.

## 2. Traditional Vaccine Development

The fundamental concept of vaccine development postulated by Louis Pasteur towards the end of the 19th century serves as the cornerstone of classical vaccination (isolation, inactivation, and injection of the causal agent). Throughout the 20th century, this paradigm drove the development of vaccines [5]. The conventional method has helped develop vaccines that have led to the eradication of diseases such as smallpox, pertussis, measles, mumps, rubella, and Haemophilus influenzae B, while generally increasing life expectancy. Based on these empirical principles, three conventional methods are employed involving the application of inactivated microorganisms, live attenuated agents, and pathogen subunits for vaccine development against different infections (Table 2) [6]. Inactivated vaccines contain severe disease-causing bacteria that have lost their capacity to infect or multiply inside or outside of a host. Such vaccines can be generated using physical, chemical, or radiation-based techniques that do not compromise microbe antigenicity. As inactivated microorganisms do not survive in the extracellular environment or within vaccinated individuals, they are generally considered safe. However, compared to other vaccine types, their application may offer reduced or short-lasting protection. Examples of such vaccines include *Brucella melitensis Rev1* (brucellosis prevention in sheep and goats), Dukoral (cholera vaccine), Shanchol (cholera vaccine), Q-Vax (*C. burnetii*), and many others (Table 2). Live attenuated vaccines are comparatively more efficient against intracellular bacteria as they contain live bacteria or viruses that have been “weakened” (attenuated) to elicit a protective immune response without inflicting disease in healthy individuals. This “weakening” of the pathogen is accomplished by inducing genetic change. Genetic changes that reduce virulence may be induced in the laboratory by repeatedly culturing the pathogen (serial passage), or by introducing the pathogen to a non-specific “foreign” host in which genetic variability and mutation cause the pathogen to lose its virulence to the natural host. Most modern vaccines contain live, attenuated pathogens; examples include the *M. bovis* strain *Bacillus* Calmette–Guerin (BCG) vaccine, vaccines against *S. enterica* spp., the *B. anthracis* vaccine (BioThrax), and the *V. cholerae* vaccine (Vaxchora) (Table 2). Subunit vaccines contain purified antigenic parts of a pathogen (instead of the whole organism) that elicit a host immune response. Purified antigens can be toxoids, subcellular fragments, or surface molecules, and they can be conveyed by a variety of carriers. Compared to live attenuated vaccines, the immune responses elicited by subunit vaccines are not as potent or long lasting. Repeated doses are often required, followed by booster doses in the following year. Subunit vaccinations frequently have adjuvants added. These are elements that support and prolong the immunological response to the vaccine. The majority of newly developed leishmaniasis vaccines target different antigens using protein subunits. Tularemia subunit vaccines have also been created in which the *F. tularensis* surface protein Tul4 and the heat shock protein DNAK were combined and delivered to mice intranasally together with the adjuvant GPI to demonstrate long-term pathogen-specific immunity [7]. Other conventional methods used for vaccine development along with their respective status are summarized in Table 2.

Conventional vaccine development strategies are often associated with various limitations such as short duration of effectiveness, numerous safety concerns, and high production costs. Inactivated and attenuated vaccine development strategies often face delays owing to the difficulties associated with the culturing of some bacterial strains. Some vaccines may not be adequately attenuated and thus may still elicit detrimental immune responses. With respect to subunit vaccines, the antigen purification process is often costly and technically challenging. While the effectiveness of traditional vaccines may be disease dependent, new diseases are also emerging due to mutations, gene exchange, interspecies transfer, and novel environmental exposures. Therefore, more reliable and novel methods are required to counter these challenges. To reduce the shortcomings in conventional vaccinology, recombinant DNA technologies have been employed for second-generation vaccine design. Highly pure antigenic components or strains that have been rationally attenuated, for example, pertussis toxins, have been developed; however, vaccines produced using this method require several years of development [8,9]. In other instances, such as the development of a universal meningococcal B vaccine, the conventional empirical approach was insufficient owing to the large number of strains and the various side effects. This issue was solved by the pan-genomic method, which compares sequences from multiple strains. Proteome of serogroup B strain MC58 was investigated through various in silico methods to find possible target vaccination antigens. The genomic method also speeds up the process of vaccine development as we see in the case of mRNA vaccines developed for COVID-19. This will help in future to develop same vaccine for intracellular bacterial pathogens. The emergence of the “genomics era” has prompted a paradigm shift in vaccine development. Genome mining coupled with functional and structural genomics research has facilitated rapid antigen identification leading to third-generation vaccine development. This review focuses on the more advanced genomics-based approaches (pathogen genome and host genome) for vaccine development to treat various intracellular bacterial pathogen. We also highlight recent advancements in synthetic biology that aid vaccine design and development to address the issue of emerging intracellular infections.

**Table 1 pharmaceutics-15-00152-t001:** Summary of intracellular bacteria, associated illnesses, host cells, and incubation period.

Bacteria	Illness	Intracellular Lifestyles	Bacterial Factors	Host Cells Localization	Incubation Period	Reference
*L. pneumophila * Non-spore forming, Gram-negative	Legionnaire’s disease	Intravacuolar pH 6	Type IV secretion system	Macrophages	2–10 days	[10]
*M. tuberculosis* Non-spore forming, Acid Fast	Tuberculosis	Intravacuolar pH 6.4	Type VII secretion system	Macrophage, Cytosol, Phagosome	4–6 weeks	[11]
*S. typhi* Rod-shaped, Gram-negative	Typhoid fever	Intravacuolar 5	Type III secretion system	Macrophages	10–14 days	[12]
*Brucella* spp. Gram-negative coccobacilli	Brucellosis, High fever	Intravacuolar 4	Type IV secretion system	Neutrophils, Vacuole	2–4 weeks	[13]
*Listeria monocytogenes* (*L. monocytogenes*) Gram-positive *Bacillus*	Listeriosis	Intracytosolic	In1A, In1B	Epithelial cells, Cytosol	1–2 weeks	[14]
*Rickettsia rickettsii* (*R. rickettsii*) Gram-indeterminant coccobacilli	Rocky Mountain spotted fever	Intravacuolar	Type IV secretion system	Monocytes/Macrophages	2–5 days	[15]
*S. flexneri* Non-spore forming, Gram-negative	Enteric disease	Intracytosolic	Type III secretion system	M Cells and Macrophages	12–96 h	[16]
*M. leprae* Acid Fast *Bacillus*	Leprosy	Intravacuolar	Type VII secretion system	Schwann cells (SCs)	4–5 years	[17]
*S. typhimurium* Non-spore forming, Gram-negative	Indigestion, food poisoning	Intravacuolar pH 5	Type III secretion system	Macrophage, Vacuole	12–72 h	[18]
*Chlamydia* spp. Gram-indeterminant	Genital and ocular infections	Intravacuolar pH ¼ 7.25	Type III secretion system	Genital epithelium and conjunctiva, Vacuole,	1–2 weeks	[19]
*S. aureus* Gram-positive	Dermal infection, osteomyelitis, mastitis	Intravascular	Type IV secretion system	Epithelial Cells, Osteoblast, Endosome, Cytosol	16–18 h	[20]
*S. enterica* Gram-negative	Paratyphoid and typhoid	Intravacuolar	Type III secretion system	Macrophage, Modified phagosome, Vacuole	12–72 h	[21]

**Table 2 pharmaceutics-15-00152-t002:** Vaccine types for immunization against different bacterial infections (reprinted from Osterloh, 2022) [22].

Vaccine Type	Vaccine (Target Bacteria)	Vaccine Status	Reference
WCA	Q-Vax (*C. burnetiid*) (Dukoral, Shanchol (*V. cholerae*) *R. rickettsia* *R. prowazekii* *O. tsutsugamushi*	Licensed Licensed Experimental Experimental Experimental	[23] [24] [25] [26] [25]
LAVs	*BCG* (*M. tuberculosis*) *S. enterica* spp. BioThrax (*B. anthracis*) LVS (*F. tularensis*) Vaxchora (*V. cholerae*) *O. tsutsugamushi* *R. prowazekii*	Licensed Licensed Licensed Licensed Licensed Experimental Experimental	[27] [28] [29] [7] [30] [31] [32]
Live recombinant bacteria	*M. tuberculosis* *R. rickettsii* *C. burnetii* *S. aureus*	Experimental Experimental Experimental Experimental	[33] [34] [35] [36]
subunit vaccines	dTAP combined vaccine (*C. tetani*) dTAP combined vaccine (*B. pertussis*) Trumenba (*N. meningitidis*) rPA102 (*B. anthracis*) *S. aureus* Trumenba (*N. meningitidis*)	Licensed Licensed Licensed clinical trial Experimental Licensed	[37] [38] [39] [40] [41] [42]
Polysaccharide conjugates	PedvaxHIB, ActHIB, HibTITER (*H. influenzae*) Prevnar 13, Pneumovax 23 (*S. pneumoniae*) Menactra, Menveo, Menomune (*N. meningitidis*)	Licensed Licensed Licensed	[43] [44,45] [46]
Viral vectors	85A antigen (*M. tuberculosis*)	Licensed	[47]
Bacterial vectors	*H. pylori* (*S. typhimurium*) encoding urease A and B subunits *Y. enterolica* encoding bacterioferritin (*B. abortus*) *L. monocytogenes* encoding antigen 85 complex and MPB7MpT51 antigen (*M. tuberculosis*)	Experimental Experimental Experimental	[48] [49] [50]
Plasmid DNA	*M. tuberculosis* (hsp65 from *M. leprae*) *B. anthracis* (PA antigen)	Experimental Experimental	[51] [52]
BGs	*Y. pestis* *S. typhimurium* enteritides (BGs expressing flagellin) BGs carrying DNA for *N. ghonorhea* antigens	Experimental Experimental Experimental	[22] [53] [54]
OMVs	Bexsero/4CMenB, VA-MENGOC-BC, MeNZB, MenBVac (*N. meningitidis serogroup B*) *B. pertussis* *BCG* *C. trachomatis* *V. cholerae* *M. smegmatis* *T. pallidum*	Licensed Experimental Experimental Experimental Experimental Experimental Experimental	[55,56,57] [22] [22] [58] [58] [58] [58]

Abbreviations: whole-cell antigen, WCA; live attenuated bactericidal vaccines, LAVs; bacterial ghosts, BGs; outer membrane vesicles, OMVs.

## 3. Genomics Revolution

The beginning of genomics can be traced to the 1970s, when DNA sequencing technology was first developed. However, the “genomics era” truly began in the late 1990s with the sequencing of the *H. influenzae* genome [59]. Thereafter, the emergence of advanced technologies made it possible to swiftly sequence a genome, thus leading to the conventionalization of whole-genome sequencing [60]. For all major human pathogens, at least one genome sequence is available. As of July 2022, 190,825 (88.2%) bacterial genomes were assigned a “completed” status, inclusive of closed genomes and whole-genome shotgun sequences, while the sequencing of 19,217 (8.9%) genomes is ongoing (https://gold.jgi.doe.gov/index (accessed on 12 August 2022). The sequencing status of the remaining genomes include 4907 (2.3%) in draft, 966 (0.4%) awaiting samples, 389 (0.2%) undergoing targeted gene sequencing, and 12,895 completed viral genomes (https://gold.jgi.doe.gov/index (accessed on 12 August 2022). The sequencing of bacterial pathogens (~4000 genes) has helped identify all antigens that can be potential drug targets. Viral pathogens have less than 10 genes; however, their genomics can be used to identify the variability between different isolates. Furthermore, the role of host genetic factors in infectious diseases is also very important. Therefore, the availability of whole human genome sequences and various ongoing human genome projects (http://www.1000genomes.org/ (accessed on 14 August 2022) are valuable assets that facilitate the identification of numerous possible vaccine and drug targets [61,62]. Genome-based approaches may potentially help identify 10–100 times more candidates in 1 to 2 years compared to conventional methods over the same period [63]. Advancements in long reading sequencing technologies will facilitate the systematic assembly of diploid genomes, revolutionizing genomics by presenting the entire range of human genetic variation, covering some unaccounted heritability, and uncovering novel disease processes. Aside from this, genome-based vaccine projects enhance our understanding of pathogenesis, epidemiology, physiology, and the function of microbial proteins [64]. Causative agents responsible for disease outbreaks can be identified by metagenomics (the evaluation of all genetic information extracted directly from a sample) [65,66]. The complete genome sequence identification of organisms serves as a starting point for the screening process of target molecules using high-throughput sequencing approaches, as shown in Table 3 [60,67]. The type of pathogen being screened determines the screening methods used for vaccine design; however, vaccine design is also based on several widely accepted principles and essential criteria for therapeutics. These criteria include the need for targets that (a) are expressed and accessible to the host immune system, or a therapeutic agent during disease; (b) are conserved genetically; (c) exhibit survival and pathogenesis significance; and (d) demonstrate no similarity to various other attributes of the host. Previous decades have seen significant advances in the application of genomic technology that has led to new vaccine development strategies against major human pathogenic bacteria such as intracellular bacterial pathogens (Figure 1) [22,68,69,70,71,72].

## 4. Genome-Based Approaches

Vaccination is an efficient and cost-effective way to control bacterial pathogenesis. Vigorous vaccination drives have saved millions of lives. However, infectious diseases continue to be the biggest cause of death globally [78]. Countless diseases still do not have adequate vaccines. Therefore, the advent of the genomic era has fundamentally altered how vaccine candidates are identified and developed. Genomic approaches have several advantages over conventional methods. Genomics may be used to discover all relevant antigens expressed uniformly in various pathogenic strains, and it is applicable to strains that can be cultured in vitro and also those that cannot. Viable but non-culturable cells (VBNCs) cannot grow on the conventional medium (agar) until appropriate conditions have been applied. Therefore, novel approaches are required to evaluate VBNCs. A novel field of molecular biology known as metagenomics (environmental and community genomics) can analyze DNA from many microorganisms. Metagenomics analysis discloses characteristics of microbial genomes and biology that had previously gone unnoticed, such as metabolic capacities, signs of host co-evolution, and possibly, community-wide selection, and the mechanisms behind cell–cell signaling that can be used for the development of novel vaccine strategies. The fact that non-culturable microbes represent the majority of organisms found in all the world’s habitats led to the establishment of metagenomics. Evidence for this came from the examination of 16S rRNA gene sequences amplified directly from the environment (a method that removed the bias introduced by culturing). This technique identified a multitude of new microbial lineages. Genomic approaches elucidate both genuine and mimetic antigens that might trigger protective immunity against bacterial epitopes [79]. Additionally, genomic data are combined with powerful technologies, including the in vivo expression technique (IVET), signature-tagged mutagenesis (STM), proteomics, DNA microarrays, two-dimensional gel electrophoresis, and mass spectrometry, to experimentally select new surface-exposed antigens or pathogenicity factors. Due to innovations in technology, such as extensive genomic, transcriptomic, and proteomic analysis, rapid target identification of novel vaccine antigens is feasible, leading to a resurgence in vaccine development. In the following section, we provide details of novel genomic approaches for future vaccine development strategies against intracellular bacterial pathogens.

## 5. An In-Silico Approach: Reverse Vaccinology (RV)

Genome mining is a major approach used in reverse vaccinology (RV). It uses the sequences of various organisms of interest, such as viruses, bacteria, or parasitic pathogens, instead of cells as starting material for the identification of new antigens that can subsequently be verified by experiments (Table 3) [80]. In this method, the pathogen is sequenced and analyzed in silico to identify genes that are most likely to encode surface-localized proteins or those that have homologies with known bacterial components involved in bacterial pathogenesis and virulence. The gene of interest (GOI) is then isolated, cloned, expressed as a recombinant protein, purified, and used in animal model testing to elucidate the capacity of the gene to confer defense against the tested pathogen (Figure 2). Alternately, in the absence of an in vivo model, in vitro approaches that elucidate vaccine efficacy in humans, such as opsonophagocytosis and serum bactericidal activity assays, may also be carried out [81,82]. Pizza and colleagues in collaboration with The Institute for Genomic Research (TIGR) first applied the RV approach to examine the intracellular bacteria belonging to *N. meningitidis* serogroup B (MenB), which is responsible for 50% of meningococcal meningitis cases worldwide [83,84]. Based on the chemical composition of polysaccharide capsules, *N. meningitidis* was identified as having 13 serogroups. Only five serogroups (A, B, C, Y, and W135) have been linked to meningococcal meningitis and sepsis. Capsular polysaccharides have been used in vaccines against serogroups A, C, Y, and W135 in both adults and infants. Interestingly, it was found that when the bacterial polysaccharide was conjugated with a carrier protein, a strong T-cell-dependent immune response was evoked following vaccination that provided long-term protection. However, such an immune response was not observed for serogroup B bacteria. Initial attempts to develop a meningococcal B vaccine by conventional methods were unsuccessful. Although there were several reasons for this failure, two primary reasons are apparent. First, meningococcal B capsular polysaccharides (CPS) (a polymer of α (2-8)-linked N-acetylneuraminic acid) had a high similarity to components of human tissues, which resulted in poor immunogenicity in humans, often stimulating autoimmune responses. Secondly, as protein-based vaccines are highly antigen-specific, they only provide defense against a very small number of strains [85]. Briefly, in RV, the virulent strain of *N. meningitides* MC58 was completely sequenced, and an in-silico method was used to identify 600 genes that encoded surface-exposed proteins [83]. Out of these 600 genes, 350 were expressed successfully in *E. coli,* and were subsequently used to examine immunogenicity in mice [83]. Enzyme-linked immune sorbent assay (ELISA), fluorescence-activated cell sorting (FACS), and immunoblot analysis were performed using the immune serum of the mice to examine surface-exposed localization [83]. These experiments identified 90 previously unknown surface-localized proteins. Bactericidal assays and/or passive protection in infant rat assays were subsequently carried out to identify 30 out of the 90 novel proteins that triggered the production of antibodies, which could eradicate the bacteria in vitro. Clinical trials were subsequently carried out using five antigens that had a high degree of conservation in multiple *N. meningitides* strains. Additionally, the five antigens could also be purified with relative ease. The findings of this study led to the development of 5CVMB, a recently released universal vaccine against *N. meningitidis*, a “cocktail” of five antigens discovered by RV (fHBP, NadA, GNA2132, GNA1030, and GNA2091) [86]. 

The RV procedure has also been used to examine many other human bacterial pathogens [83]. The antigens that were identified have been successfully advanced to the clinical and development phases [87,88,89,90,91,92]. The list of bacteria includes *Streptococcus* species (*S. agalactiae, S. pyogenes, S. pneumoniae*), *B. anthracis*, *Porphyromonas gingivalis* (*P. gingivalis*), *M. tuberculosis*, *H. pylori*, and *Chlamydia pneumoniae* (*C. pneumoniae*) (Table 4) [87,88,89,90,91,93,94]. Interestingly, the investigation of 8 group B *streptococcus* genomes led to the discovery of 312 surface proteins, of which 4 proteins progressed to vaccine development and were later found to be effective against all subtypes [95]. For group A *Streptococcus* (GAS) bacteria, a vaccine containing a small number of bacterial proteins has also been produced [96]. In both instances, the RV approach was crucial for ensuring that the chosen antigens had no similarity to proteins encoded by the human genome. The sharing of epitopes between host antigens and streptococcal bacteria is referred to as molecular mimicry [97]. Damian was the first to propose the word “molecular mimicry’ from an evolutionary perspective [98]. He postulated that antigenic sharing between microbes and host tissue could be seen as a way for pathogens to evade the host’s immune response [98]. In this context, the very first example was provided by Zabriskie in GAS [99]. The hallmark of rheumatic fever pathogenesis is molecular mimicry, where the GAS carbohydrate epitope, N-acetylglucosamine, and *streptococcal* M protein structurally mimic cardiac myosin in human disease [100]. In animal models immunized with the M protein of streptococcus and cardiac myosin, pancarditis was observed [100]. GAS vaccine candidate screening is based on the conserved M epitope, N terminal M peptides, the cell surface, and secreted proteins [101]. Genetics and proteomics can be used to precisely formulate the antigen/epitope composition of novel vaccines. However, a vaccine is still not available, and various options are being examined for vaccine development. Vaccines based on the M protein (purified M protein, StreptInCor, 30 valent HVR StreptAnova, J8/J14/p145) are in phase I clinical trials, while the 23 valent HVR vaccine candidate has moved on to clinical trial phase II [101]. Comparative genomics is another approach that can be used to identify possible candidate GAS vaccines.

Protein-based vaccines are also being developed using a genome-based strategy to protect against emerging infectious diseases caused by *Streptococcus pneumoniae* (*S*. *pneumoniae*) and antibiotic-resistant *S. aureus, C. pneumoniae*, etc. Additionally, a vaccine against *Chlamydia* has been developed using this strategy [102]. *Chlamydia pneumoniae* is a pneumonia-causing species of intracellular bacteria. The absence of an effective pneumonia vaccine has allowed the bacteria to develop substantial antibiotic resistance. Researchers have attempted to utilize proteomics and RV techniques with the aim of developing a multi-epitope vaccine [103]. The efficacy of the vaccine was validated using immunology and bioinformatics-based integrative pipeline (a series of software algorithms to generate processes and data) approaches. The vaccine developed by this method exhibited sustained binding to TLR4, MHCI, and MHCII receptors, as well as the capacity to trigger the host immune response. Further scientific evaluation of vaccine candidates will enable us to efficiently combat endemic diseases caused by intracellular bacterial pathogens. Most vaccines designed using the RV approach are in the discovery phase, while some candidates have reached phase I and II clinical trials (Table 4). The main disadvantages associated with protein-based vaccines include low protein abundance and/or solubility. Moreover, proteins that are only expressed in vivo may also be difficult to identify. 

The RV method has also been utilized to examine *M. tuberculosis,* a bacterium which causes tuberculosis (TB), the thirteenth leading cause of death (1.6 million) and second leading infectious killer (COVID-19 being the first) worldwide (https://www.who.int/news-room/fact-sheets/detail/tuberculosis (accessed on 15 November 2022). Low efficacy of the *Bacillus* Calmette–Guérin (BCG) vaccine against pulmonary TB, coupled with the emergence of multidrug-resistant TB, warrants the need for novel TB vaccines. The BCG vaccine is not universally approved. It is not administered in many developed countries; however, it is still used in underdeveloped and developing countries. The status of TB in different countries, in relation to BCG vaccination is available online (http://www.bcgatlas.org/ (accessed on 16 November 2022). Anti-mycobacterium vaccines are urgently required. With respect to this, several web-based RV programs (MycobacRv, Violin, VaxiJen, and MtbVeb) have been developed to aid the scientific community. MycobacRv is a database of potential mycobacterial adhesin vaccine candidates identified from 22 mycobacterial strains. The database also provides detailed analysis of predicted adhesins/adhesin-like and extracellular/surface-localized proteins, which may help in the creation of epitope-based mycobacterial vaccines [104]. Another web-based database called the Vaccine Investigation and Online Information Network (VIOLIN) combines mined vaccine literature with data curation and storage. Additionally, it offers a framework that can predict possible targets for vaccines against different pathogenic invaders [105]. Recently a study analyzed the proteome of *M. tuberculosis* H37Rv using a prediction software called the New Enhanced Reverse Vaccinology Environment (NERVE) to identify vaccine targets. NERVE identified 331 proteins, which were then further analyzed by VaxiJen software (Figure 3). Specific filter parameters (adhesin probability value 50%, antigenicity value ≥0.5, and no homology to human proteins or transmembrane sections) were used to identify 73 antigens. Further search refinements involving detailed literature and protein analysis revealed six novel vaccine candidates (EsxL, PE26, PPE65, PE_PGRS49, PBP1, and Erp), which were subsequently approved for TB vaccine development [106]. Clinical trial data for PE_PGRS49 (a DNA vaccine with a protein-related member Rv1818c) have been reported [107], while data for the remaining five have not. The inclusion of PE_PGRS49 in existing BCG vaccines may enhance their efficacy [106]. Recently, the RV strategy was successfully applied to many other intracellular bacterial pathogens for vaccine development, including *Brucella* spp., *C. pneumoniae*, *R. prowazekii*, *Anaplasma* spp., *Ehrlichia* spp., *S. pyogenes*, etc., [25,103,108,109,110,111,112,113,114]. 

The fundamental limitation of the RV strategy is that it only examines one single strain within a species, and does not address issues that may arise owing to genetic diversity. This risk became apparent when a comparison of genome sequences from several strains of *S. agalactiae* were carried out. Eighty percent of the *S. agalactiae* core genome is composed of genes, with each new genome displaying ~18% variation. This finding served as inspiration for the development of a “universal anti-*S. agalactiae*” vaccine. This vaccine was composed of four antigens, with one antigen being present in all strains, and this combination provided protection against all strains. The evolution of sequencing approaches provided researchers with an enriched bacterial genome sequence. This opened a new area of vaccine development known as pan-genomics. The comparative or pan-genomic method was viewed as the development of classical RV. This method facilitated the development of universal vaccines based on a core genome shared by all strains, which would be effective against various strains of the same species [115,116,117,118]. 

## 6. Pan-Genomics Analysis or Comparative Genomics

Following publication of the first bacterial genome sequence, technological advancements have led to the elucidation of numerous complete genome sequences. The genomic databases now have more than 250,000 bacterial genomes, a result of intensive work and more efficient sequencing methods [119]. The immense genomic variation among individuals of the same species is one of the most profound discoveries that have been made. Many mechanisms govern genetic diversity, such as genetic drift, natural selection, and gene flow due to mobile elements, transposons, and non-random mating [120]. Subtractive hybridization and comparative genome hybridization (CGH) techniques have shown that intraspecies genetic diversity is as significant as interspecies diversity [121] Genomic comparisons of multiple isolates of bacterial pathogens and closely linked pathogens are important for the scientific community, as knowledge regarding genome size, gene content, conservation, and gene diversity in various strains and disease conditions has profound implications on vaccine development. Further improvements in sequencing technology facilitate comparisons among multiple complete genomes. Pan-genomics (an advancement of classical reverse vaccinology) is utilized to make comparisons amongst multiple strains of bacteria. 

The term “pan-genome” was first used by Tettelin et al. in 2005 (Figure 1) [122]. It was explained as the entire gene repertoire of a particular species. By comparing the complete genomes of eight different strains of *S. agalactiae* (group B *streptococcus*; GBS), representing the genetic variety of the species, intraspecies heterogeneity was demonstrated. The first application of pan-genomics was carried out to design a universal vaccine against GBS [114]. Computational techniques were used to predict 589 genes encoding surface-associated proteins, 396 of which were core genes, while the remaining 193 genes were lacking in at least one strain. Potential antigens were subsequently selected and expressed as recombinant proteins. These recombinant proteins were purified and their anti-GBS potential was evaluated. Only four of the candidate antigens elicited immunity in animal models. Out of these four, one was a component of the core genome; however, the antigen was unable to provide universal protection against GBS. Therefore, a combination of the four antigens should be included in the final vaccine formulation. This example clearly illustrates the need to understand pan-genomics. Phase I testing for a GBS vaccination is currently underway (http://clinicaltrials.gov/ct2/show/NCT01193920 (accessed on 17 November 2022). For GAS, comparative genomics was employed to search for a global vaccine candidate using 2083 globally sampled GAS genomes [123]. The study reported 290 related genomic phylogroups from 22 countries, emphasizing the difficulties of developing vaccines that have universal applicability.

Machine learning has facilitated the pan-genomic sequencing of 1595 strains of intracellular *M. tuberculosis* (Mtb) bacteria [124]. Amongst these strains, 946 bacteria were resistant to anti-tuberculosis medicine (isoniazid and rifampicin) [124]. A pan-genomic analysis of 36 intracellular Mtb isolates identified 67 super core genes (SCGs), of which 28 were very important and reflected phenotypic generality [105]. Most SCGs have been shown to play significant roles in Mtb pathogenicity and code proline-glutamate/proline-proline-glutamate (PE/PPE), virulence factors (VFs), antigens, and transposases [125,126,127]. The proteins are the multifunctional immune modulators. PPE19 is highly expressed in macrophages (two copies/Mtb strain), which promotes Mtb intracellular survival [127]. Another SCG, phospholipase C (plcC; three copies per Mtb strain) aids Mtb escape from phagosomal vacuoles and hence degradation [126]. The study carried out a pan-genome investigation of Mtb to determine which genome was primary, secondary, phenotype-general, phenotype-specific, and interconvert during evolution [126]. This study provides a novel research paradigm for investigating organisms using a pan-genomic approach as well as laying an important theoretical foundation for TB research.

In a recent study, a silicon-based multi-epitope vaccine was designed based on 200 Mtb genomes [128]. Two vaccine candidates, ESA-6-like proteins and diacylglycerol acyltransferase were generated from multi-epitope mapping. These two vaccines exhibited rigorous van der Waals and electrostatic binding forces, stable dynamics, and high binding affinities for several immunological receptors. The researchers noticed remarkable primary, secondary, and tertiary immune responses to the antigens in addition to a raised interleukin and interferon count [129]. In conclusion, the proposed vaccines appear to be viable candidates for further evaluation. Their true biological value as agents that prevent drug-resistant *M. tuberculosis* infections must be determined, along with their respective modes of action.

*Salmonella typhi*, *S. typhimurium*, and *S. enterica* cause typhoid, paratyphoid fever, and gastroenteritis. Novel antimicrobial agents are urgently needed to treat these conditions. Pan-genome analysis estimates that *Salmonella* has an “open” pan-genome that has 10,775 gene families. There are 2847 core gene families (CGFs), 4657 dispensable gene families (DGFs), and 3271 strain-specific gene families (SSGFs). The study concluded that essential core gene families (E-CGFs) may serve as important targets for the development of novel antimicrobial drugs [130]. Another pan-genome study, conducted in Brazil, revealed that while the pan-genomes of *S. typhimurium* under study were open, they especially tended to close for ST313 strains [131]. 

*Legionella pneumophila*, an intracellular bacterium linked to Legionnaire’s disease, was found to infect ~90% of patients [132,133,134]. *Legionella pneumophila* is resistant to many drugs. Genomic approaches have been considered for the development of new vaccines. In a recent study, subtractive proteomics (pathogen genes necessary for its survival, yet absent in the host, are “subtracted”) and immunoinformatic tools were used to design a highly immunogenic vaccine containing five proteins (Q5ZVG4, Q5ZRZ1, Q5ZWE6, Q5ZT09, and Q5ZUZ8) against *L. pneumophila*. The proposed vaccine elicited a host immune response against *L. pneumophila* [134]. However, to demonstrate the protective immunological efficacy of the vaccine, experiment-based studies are strongly advised. The vaccine should be applied to animal models followed by human clinical trials. *Rickettsia rickettsii* (an intracellular coccobacillus bacterium) causes Rocky Mountain spotted fever (RMSF). Docking analysis, reverse vaccinology for protein identification, and pan-genomics were used to analyze 47 *Rickettsia* genomes. Researchers found 90–100% homology among the four *Rickettsia* groups. In addition, eight protein types were also identified in this research to support polyvalent vaccine development against *Rickettsia*, while nine candidates were identified as drug targets [135]. Although the study predicted that polyvalent vaccines would interact with the majority of microorganisms in this vast group, potentially leading to a vaccine, predictive data must also be supported with in vivo and in vitro data.

Shigellosis is a disease commonly found in children from developing countries. It is caused by the *Shigella flexneri* (*S. flexneri*) bacterium. Researchers have compared the virulence signatures and genomic characteristics of the of *S. flexneri* stereotype 3b genome, SFL 1520, with commonly found genomes of *S. flexneri* to identify a link for vaccine development [136]. The study reported that SFL1520 shared significant similarities with other *S. flexneri* serotypes in the phylogenetic analysis based on core genes, but there were also notable differences in SFL1520’s accessory genes. The discovery of a substantial number of distinct genes in SFL1520 suggests extensive horizontal gene acquisition in a relatively short amount of time. Major virulence characteristics of SFL1520, such as serotype conversion and multidrug resistance, were created by these acquired genes, which will further help in vaccine design. A deeper understanding of strain variability could serve as a foundation for learning more about variations in pathogenesis and pathogen–host interactions that can translate to healthcare benefits. The *Chlamydia* group consists of a single genus, containing the species *Chlamidia trachomatis* (*C. trachomatis*)*,* a pathogenic intracellular microorganism involved in sexually transmitted diseases causing “genetic infection” [137]. In a pan-genomic analysis of 16 genomes examining a combination of core genomes (small and large), the stability of *Chlamydia* genomes was verified, and various distinct genome characteristics were discovered [138]. The presence of a conserved core genome (large) and evolvable variable (small) genome balance the selective pressure towards genome reduction and the requirement for adaptation to evade host immunity. 

Gastritis and stomach ulcers are caused by the Gram-negative bacterium *H. pylori*. One to two percent of infected individuals may develop stomach cancer. *Helicobacter pylori* can be transmitted from person to person through saliva, or through contaminated food or water. Additionally, 80% of infected people have no symptoms, and bacterial resistance to medication has increased over the last decade. To control *H. pylori* infections, vaccine-based prevention has been prioritized rather than subsequent antibiotic treatment. The RV strategy in combination with pan-genomic analysis was carried out in a study to analyze 39 *H. pylori* isolates [139]. A total of 28 non-host homologous proteins were identified as common therapeutic targets for future vaccine development [139]. Table 5 shows the details of antigens or approved vaccines against bacterial pathogens that were identified by the pan-genomics RV method.

## 7. Antigen Prioritization

Another primary disadvantage of the RV method is that it is often expensive and labor-intensive. High-throughput screening of countless antigens also makes this approach time-consuming. RV findings have demonstrated that when seeking new vaccine candidates, surface-exposed bacterial proteins that may trigger a strong immune response should be taken into special consideration. More stringent in silico screening measures and the application of new experiment methodologies should be prioritized to meet these requirements. A study examining the relationship between Tourette syndrome (TS) (a neurodevelopmental disorder beginning in childhood or adolescence) and *S. pyogenes* (or Group A streptococcus (GAS) antigen, including subsets of M proteins, streptolysin O (SLO), streptokinase A, and C5a peptidase precursors)-induced immune response implemented this strategy [118]. Many studies have reported a highly evolved relationship between TS and streptococcal infection [147,148,149]. Stringent in silico analysis was used to filter gene sequences to predict surface-associated proteins that had canonical motif-associated surface-exposed proteins. These surface-exposed proteins contained a lipoprotein signature, host cell binding domains (RGD), leader peptides, and outer membrane anchoring motifs. The screened protein coding genes were cloned, purified, and used for GAS-specific protein microarray analysis. The detailed protocol has also been reported [118]. The array proteins were then further examined using human sera from asymptomatic isolates, symptomatic isolates of pharyngitis, and TS isolates. This provided the first proof that the sera from TS patients had immunological profiles similar to patients who exhibit a strong, specific, and broad immune response. The same strategy may be applied to identify novel vaccine candidates. This in silico method of canonical surface motif screening minimizes the number of genes that have to be cloned and the number of proteins that have to be purified, ultimately reducing the quantity of pathogenic antigens that need to be examined to identify high antigenicity. 

Another approach prioritizes candidate antigens for vaccine development based on immunogenicity. This approach has been used with respect to several intracellular bacterial pathogens [146,150,151]. In this method, extensive libraries of peptides expressed on the surface of *E. coli* are created and analyzed using human serum from infected or recuperating patients. The main advantage of this approach is that it is a simple serum analysis method; however, a disadvantage is that the proteins are not expressed because they do not fold properly. Although in this process, some antigens might be overlooked when comprehensive cloning, expression, and purification are not possible, it is both cost- and time-efficient. The proteomics-based strategy, which entails the proteolytic “shaving” of live bacterial cells (to identify proteins that are easily accessible to host immune functions), is another experimental means to prioritize specific antigens from a list of in silico selected surface proteins. Briefly, proteases are used to selectively digest bulging proteins from whole cells that have been identified by the mass spectrometry analysis of released peptides and genome-based antigen analysis. Rodriguez-Ortega et al. also used the same method for GAS surface proteome analysis and identified 70 proteins [146]. Known protective antigens were excluded from the selection, while new antigens were expressed, purified, and later used in mice immunization experiments. This technique has also been carried out to examine a wide range of bacterial pathogens, including M. *tuberculosis*, S. *dysenteriae*, *F. tularens*, and *B. anthracis* [152,153,154,155,156,157].

## 8. Functional Genomics: Genes Essential for Vaccine Candidates

Understanding pathogenesis and the relationship between pathogens and hosts depends on bacterial gene expression and function. An illustration of the role of genomics in accelerating the discovery of promising vaccine candidates is the identification of pili (long filamentous appendages expanded from bacterial cell surface) in key pathophysiological strains of streptococci. This was made possible by the discovery of distinctive gene organization in pilus islands [143]. Pan-genomic RV at Novartis Vaccines revealed three protective antigens of GBS that assemble into high-molecular-weight polymers that can be observed by electron microscopy as pilus-like structures [158]. Additionally, three pilus islands that encode various structural pilus types have been found in GBS. Each pilus has two antigens that can trigger immune responses in mice [159]. The pilus subunits of *S. pneumoniae* are also immunogenic in mice [159]. They are also effective at offering safety in both active and passive immunization paradigms [159]. Active immunization uses an immunogen to trigger a host response, leading to the production of antibodies against the specific disease. Passive immunization involves directly administering pre-prepared antibodies into a host. Functional and structural genomics were used to investigate bacterial gene expression and pathogen–host relationship.

The discipline of functional genomics has emerged from molecular biology. It is characterized by advancements in cutting-edge technologies such as transcriptomics and proteomics. In addition to genetic content, these technologies also investigate transcription and expression profiles, so that suitable vaccine candidates may be identified and developed (Figure 1).

## 9. Transcriptomics: Expression Profile Identifies Potential Vaccines

Transcriptomics offers a comprehensive perspective of a pathogen’s full transcriptional activity and allows the comparison of gene (RNA transcript) expression under various growth and environmental settings. Knowing which genes are overexpressed in vivo during infection is crucial for the discovery of vaccine antigens, potentially serving as candidates for prospective vaccines. Owing to whole-genome sequencing, complete microbial genome sequences are now more easily accessible, which has facilitated their usage in various vaccine development strategies. 

To identify promising MenB vaccine candidates, Grifantini and colleagues conducted microarray-based transcriptional profiling of the intracellular bacterium, *N. meningitidis* [160]. In this investigation, bacteria were cultured with human epithelial cells. Adhered bacteria were retrieved, and total RNA was extracted over a period. RNA was also extracted from non-adherent bacteria that were cultured in the absence of epithelial cells. DNA microarrays that contained the complete repertoire of the MenB gene amplified by PCR were compared. Twelve proteins that were found to be highly stimulated during adhesion were purified. The purified proteins were subsequently utilized to generate antisera in mice. A total of five sera demonstrated antibactericidal properties. Furthermore, subsequent transcriptomic investigations revealed that 48 differentially expressed genes (DEGs) were also present during the interaction of *N. meningitidis* with blood–brain barrier endothelial cells [161]. Of the 48 genes, 41 were overexpressed, while 7 were downregulated. The majority of DEGs were involved in metabolism, transport, protein production, and pathogenicity. Transcriptional profiling of MenB has identified genes that were predicted to encode for pathogenesis-related proteins [160]. In these studies, various experiment conditions, such as human serum and endothelial cell exposure, iron restriction, oxygen starvation, etc., were applied [160,162]. Several new genes that are likely to encode proteins implicated in the disease process were identified, along with their potential function. For example, the ferric uptake regulator (Fur) and iron-regulated operon of *N. meningitidis* are necessary for oxidative stress defense. Furthermore, fumarate and nitrate reductase regulator (FNR) proteins and FNR-associated proteins carry out carbohydrate fermentation, which is necessary for MenB survival in regions of the host where oxygen is limited.

The transcriptomic comparison of a clinical isolate of *S. aureus*, UAMS-1, with a laboratory strain, RN6390, revealed significant differences in the expression of genes encoding surface proteins (elevated in UAMS-1) and genes encoding proteins involved in exotoxin production (low in UAMS-1, having exotoxin 2 or 3), shedding light on pathogenesis, and emphasizing the value of investigating clinically appropriate strains for vaccine development [163,164]. With the discovery of many advanced sequencing technologies (RNA sequencing), the complete transcriptome of pathogenic bacteria, containing complete gene expression data, can be harnessed by researchers. The Kyoto Encyclopedia of Genes and Genomes (KEGG) pathway analysis tool enables systematic analysis of gene function, which may shed light on bacterial pathogenesis. Transcriptomic profiling of *M. tuberculosis* in an early tuberculosis BALB/c and SCIF immunocompetent mouse model showed that the activation of 67 genes in mice lungs correlated with the activation of the host immune response [165,166]. The benefits of transcriptome analysis for vaccine development greatly increased following technological advancements in the in vivo isolation of microbial RNA from tissues [167,168]. 

The rapid sequencing of cDNA and quantification of sequence reads, made possible by the increasing availability of microarrays (for example, for 39 different pathogens arrays are available freely from J. Craig Venter Institute: Pathogen Functional Genomics Resource Center; http://pfgrc.jcvi.org/index.php/microarray/available microarrays.html (accessed on 25 September 2022), should allow for further transcriptome-based vaccine development [169,170].

The use of next-generation sequencing (NGS) to investigate complex processes, such as bacterial infections and diseases, has greatly advanced our understanding of crucial components at both the genomic and transcriptome level, giving light to the changes within pathogens in response to therapeutic interventions [171,172]. For example, NGS has been used to identify the mutation and recombination events that enabled 240 multidrug-resistant strains of *S. pneumoniae* to evolve [171]. A phylogenetic reconstruction of the genesis and global distribution of these strains was made possible by the identification of more than 700 recombination events and 57,736 single-nucleotide polymorphisms (SNPs) in total. Similarly, use of RNA-Seq (to measure the complete transcriptome quantitively) to analyze the transcriptome of *H. pylori* RNA, revealed a multitude of complexities, and the detection of hundreds of transcriptional start sites [173]. These state-of-the-art technologies have helped scientists obtain sequencing information of bacteria in a short period of time, facilitating rapid vaccine design. 

## 10. Proteomics: A Genomic Complement for Vaccine Development

Another functional genomics tool is proteomics, the examination of proteins expressed in a cell. It enables the varied expression of proteins to be characterized as well as their location; for example, proteins that are found outside the cell, that is, on the cell surface proteome (surfaceome), are essential for triggering immune responses [174,175]. Information processing (mediated by proteins on the cell surface) is necessary for communication between immune system cells and the organism. The cell surface proteome is composed of a collection of functionally diverse proteins that both promote and inhibit the ability of cells to interact with one another in the environment. Proteomics-based methods for identifying surface-associated immunogenic proteins, which can serve as vaccine candidates, have rapidly advanced in the genomic era, and are now commonly regarded as efficient technologies that complement traditional genomic methods. The availability of an ever-increasing number of whole-genome sequences has enabled the proteomic community to identify proteins of interest swiftly and accurately from various cell compartments. In order to overcome the limitations of genomics methodologies, such as the fact that (1) expression levels of mRNA do not accurately reflect the quantity of active protein in a cell, (2) gene sequences provide insufficient data on post-translational alterations, and (3) dynamic cellular processes are not defined by genomic information, proteomics has been employed in a variety of ways to uncover new vaccination candidates against a variety of human illnesses [67,176,177,178,179,180,181]. For the proteomics approach, two-dimensional gel electrophoresis or liquid chromatography (LC) are used to reduce the complexity of samples by separating proteins or peptides for analysis by mass spectrometry (MS). The primary tool used for protein identification and characterization is MS, which has improved significantly in the past decade [182]. There are three components to MS: ion sources, mass analyzers, and ion detection systems. For the analysis of proteins through MS, three parameters are considered: (1) a protein’s ability to ionize and produce gas-phase ions, (b) the separation of ions based on their mass to charge ratio, and (c) ion detection [183]. Tandem mass spectrometry (MS/MS) and matrix-assisted laser desorption/ionization time-of-flight (MALDI-TOF) were used to determine each peptide’s molecular mass [67]. Peptide mass fingerprinting is a high-throughput protein identification method. The unknown protein is digested (with enzymes into smaller peptides), analyzed, and then the peak list of peptides in the unknown proteins is compared to a theoretical list of the protein mass database for identification [184,185]. Detailed MS procedures have previously been reported [186]. However, this approach does not effectively identify hydrophobic proteins. Tandem mass spectrometry and two-dimensional liquid chromatography (2D-LC-MS/MS) were subsequently developed and demonstrated to be very helpful in the identification of extremely hydrophobic or basic proteins that were insufficiently expressed and had large molecular weights and excessive isoelectric points [187,188]. 

Many important methods have been developed to identify surfaceomes that have minimal cytoplasmic protein contamination. The surfaces of live bacteria are carefully digested with proteases using mass spectrometry analysis as a guiding tool. This method was also applied to bacterial culture supernatants to distinguish and analyze the secretome of bacteria [189,190]. Proteomic studies have been utilized to understand how the environment influences the pathophysiology of various microbes in addition to examining microbe–host interactions [91]. With the passage of time, there has been significant advances in proteomics approaches for the screening of vaccines. A comparative approach examining differential expression parameters, such as non-virulent strains versus virulent strains, less invasive strains versus more invasive strains, or colonizing strains versus non- colonizing strains. 

The most promising vaccine options for most bacterial infections are proteins that are either secreted or surface-exposed and can induce a protective immune response. In silico analyses of surface-associated proteins predict that they account for 30–40% of bacterial proteins. Rodriguez-Ortega et al. developed a novel proteomics-based method involving the precise isolation of bacterial surface proteins [146,152]. The proteomics approach was applied to *C. pneumoniae*, an obligate intracellular bacterium causing respiratory infections. *Chlamydia pneumoniae* also contributes to atherosclerosis and heart disease. Molecular characterization of the chlamydial cell surface is still limited due to the inherent difficulties of working with *C. pneumoniae* and the lack of reliable methods for its genetic manipulation, leaving the processes of entry largely unexplained. Therefore, a genomic and proteomic approach was used to elucidate the organization of surface proteins. This was based on various factors, including predicative data from the reported genome, peripherally located proteins, heterologous protein expression, selected protein purification, mouse immune sera production, immunoblotting, FACS analysis (for surface antigen identification), and two-dimensional electrophoresis (2DE) [91]. FACS-positive antigens were identified in the *chlamydial* cell by mass spectrometry analysis of 2DE maps of protein extracts. The results showed that 53 positive FACS sera were identified. Out of these, immunoblotting detected 41 proteins of the correct size. Furthermore, 28 surface-exposed proteins that have potential as vaccine candidates were also identified by 2DE maps of protein extracts [91]. The study was the first systematic attempt to explain the organization of surface proteins in *C. pneumoniae*. 

Several other bacteria have also been examined using the proteomics approach. For example, *Salmonella typhimurium* (*Salmonella* serotype (ser.) *typhimurium*), a Gram-negative bacterium, has been well characterized by proteomic analysis [191]. In 2006, analysis of the *Salmonella* proteome (∼300 proteins) was performed using infected RAW 264.7 macrophage cells [192]. Cell lysis and differential centrifugation techniques identified an excess of 2000 bacterial proteins [192,193]. The generated extensive dataset of the *Salmonella* proteome showed substantial adaptations of the intracellular pathogen to host epithelial cells, including metabolic remodeling and differential control of virulence factors. Using the proteomics data of intracellular bacteria, the characterization of mutants can also be performed. For instance, to examine the functional roles of *Salmonella*-induced filaments, the Hensel group examined the intracellular proteome of two *Salmonella* mutant strains, *ΔssaV* (secretion system apparatus protein, ssaV) and *ΔsseF*. Furthermore, the proteome of *Salmonella* mutants deficient in *ydcR* (a known regulatory gene that controls the expression of virulence factor *SrfN*) was also summarized [192]. The proteomes of many other intracellular bacteria, including *S. flexneri*, *L. monocytogenes*, *B. pertussis*, *Brucella* (*B*.) *abortus*, *R. prowazekii*, etc., were also investigated for vaccine development by many groups [194,195,196,197,198]. This research opens the door to the development of a potent vaccine and offers a broad strategy for the design and development of vaccines against other intracellular diseases.

## 11. Next-Generation Epitope Mapping and Vaccine Design: Structural Genomics/Vaccinology

The increasing availability of genomic sequence data led to the development of structural genomics. Structural genomics involves the advanced application of many structural biology technologies, such as X-ray crystallography, nuclear magnetic resonance (NMR), spectroscopy, and electron microscopy (EM), which are increasingly being used for vaccine development (structural vaccinology, SV) [5]. SV is a genome-based strategy concerning the structural elucidation of immunodominant and immunosilent antigens, and the data can be used to design and develop peptide analogs of bactericidal epitopes [199,200]. SV has enabled antigen optimization and the large-scale industrial production of various combinations of antigens, which enforce a higher degree of immunogenicity, improved safety profiles, and enhanced protection [201]. 

The advantages of the SV approach have started to become more prominent. There are numerous examples where SV has facilitated the creation of vaccines that have previously inconceivable antigen combinations. Here, we go over a few of these instances and how structural vaccinology has helped us better understand the protective epitopes of major intracellular bacterial pathogens. Capsular polysaccharides conjugated to a carrier protein are used in approved vaccines against *N. meningitidis* serogroups (A, C, Y, and W135), but cannot be used against MenB, as the capsular polysaccharide is chemically identical to a human self-antigen. Recently, antigens have been identified that could possibly be used to develop vaccines against MenB. A surface-exposed lipoprotein, factor H-binding protein (fHbp), is one such protein antigen [202,203,204,205]. Although this antigen has more than 500 documented variants in its amino acid sequence, it is rather effective at eliciting protective antibodies. A study involving in vitro bactericidal assays (in which antibodies mediated the complement-dependent killing of bacteria) suggested that such variants could be divided into two or three different variant groups (referred to as variants 1, 2, and 3, henceforth) which do not elicit cross-protective immunity to one another [202,204]. Therefore, antigens that have specificity for all variants should be included in the vaccine design process. Although this would lead to increased complexity and an expensive production process, it would be valuable to have a single antigen that can trigger protection against each fHbp sequence variant. This was made possible by structure-based design, where epitopes from each fHbp antigenic-variant groups were designed into a single molecule. NMR and X-ray crystallography were used to identify the three-dimensional structure of fHbp, which revealed a core structure made up of two barrels joined by a shorter linker [206,207,208]. The epitopes against the three antigenic variants were recognized by protective monoclonal antibodies and later identified [202,209]. Through a series of verification experiments, 54 molecules were designed expressed and purified. Mice were subsequently immunized with the molecules, and their sera were examined for bactericidal antibodies against MenB strains possessing fHbp 1,2, and 3 variants. Many molecules triggered a significant immune response. One molecule with high activity was chosen for additional investigation, which later demonstrated the ability to induce bactericidal antibodies against MenB strains containing fHbp variants 1, 2, and 3. The crystal structure analysis of the molecule revealed perfect conservation of the original fold, making it an excellent contender for the upcoming meningococcal vaccinations.

The SV approach has also been used for next-generation vaccine development against *Bacillus anthracis*, based on its virulent and secretory proteins [210]. Five virulent proteins have been examined in *B. anthracis* to find immunogenic epitopes and develop a multi-epitope vaccine. The adjuvant Toll-like receptor 4 (TLR4), cytotoxic T lymphocytes (CTLs), helper T lymphocytes (HTLs), B cell epitopes, and linkers were added to 24 distinct subunit vaccines in various permutations and patterns. The study screened for a vaccine candidate by employing a novel strategy of SV to combat *B. anthracis*, which was previously regarded as a dangerous bioterrorism agent [210]. For the past 20 years, Willmann’s group at EMBL Hamburg has been analyzing mycobacterial proteins, utilizing high-resolution structural biology [211]. They elucidated the cryo-EM (cryogenic electron microscopy) structure of the mycobacterial *SX*-*5* type VII secretion system ‘ESX-5′ protein complex, a key component governing the virulence of mycobacterial pathogens. The structural data will aid in future vaccine design and development, as the protein complexes are a potential therapeutic target.

Structural genomics/vaccinology provides detailed insights into the structure of proteins, allowing scientists to fully consider various functional attributes. Structural genomics also provides high-quality three-dimensional structures and allows protein structure libraries to be generated, providing valuable information about wild type and mutated proteins. Structural genomics helps scientists study the molecular interaction of proteins and their folding patterns, which may further facilitate novel therapeutics. It can advance our knowledge of immune recognition mechanisms, and aid in the rational design of target epitopes that could be exploited as vaccine candidates. However, the practical applications of SV still need to be fully realized. The method cannot be accurately applied to proteins that have a low (>30%) sequence resemblance. While determining unique folds for sequences that differ from those in the Protein Data Bank, de novo protein structure prediction must be implemented. A majority of researchers who study structural genomics examine individual protein domains or subunits rather than the complete or complex protein. 

## 12. Synthetic Genomics: The Future of Vaccine Development

Synthetic genomics is a nascent domain of synthetic biology concerned with the creation of living organisms using genetic material. In this field, genes, chromosomes, gene networks, and whole genomes are combined with the help of computational approaches for chemical DNA synthesis. Synthetic genomics aims to create new genomes that can code for specialized cells with desired traits by making significant modifications to chromosome DNA, packaging, and then introducing the material into an organism.

The history of synthetic genomics dates to 2010, when a synthetic organism was created for the first time by researchers. The J. Craig Venter team presented a multistep procedure to assemble the entire *Mycoplasma genitalium* genome. This development was further enhanced by Synthetic Genomics Inc., a company that focuses on the study and commercialization of specifically constructed genomes [212]. Synthetic genomics, which includes novel protein designs, structural analysis, and the production of recombinant vaccines for quickly evolving microbial infections, has the potential to accelerate vaccine development. David Willetts’ well-known adage, “Engineering biology to cure us, feed us, and fuel us”, is most appropriate for synthetic biology because it can affect almost every aspect of our life. The need and demand for novel innovative drugs has increased due to drug resistance. 

Tuberculosis (TB), one of the deadliest risks to human health, and has pandemic potential (https://www.tballiance.org/why-new-tb-drugs/global-pandemic (accessed on 18 November 2022). Synthetic Genomics Inc. has developed a drug against TB. The bacterium *Actinobacterium mediterranei* (which produces rifamycin B and has a gene cluster for rifamycin polyketide synthase) was modified by the genetic-synthetic method to generate mutant strains that can produce 24-desmethylrifamycin B (a rifamycin analogs). The analog addresses multidrug resistance (MDR) issues involving *M. tuberculosis*. Interestingly, 24-desmethylrifamycin B derivatives, namely, 24-desmethylrifampicin and 24-desmethylrifamycin, showed 10 times greater activity than rifampicin [213]. The synthetic genomics approach may also be used to generate drug analogs that can be used to treat other dangerous types of MDR TB, such as extensively drug-resistant TB (XDR-TB) and totally drug-resistant TB (TDR-TB) [214].

## 13. Conclusions and Future Direction

Through the course of human history, there have been multiple pandemics of infectious diseases, which have caused people’s viewpoints to change. The advent of the genomic era has witnessed a change in the paradigm of vaccine design from traditional culture-based approaches to high-throughput genome-based approaches. There are demands for proactive and collaborative worldwide research and vaccine development programs. To counter some of the deadlier, more widespread infectious pathogens, such as those that cause TB, meningitis, malaria, AIDS, and dengue, vaccines need to be researched and developed on an urgent basis. Where all other conventional methods fail, genomics offers an opportunity for vaccine development against widely dispersed emerging pathogens.

The genome-based revolution in vaccine development has made information regarding a pathogen’s genome, transcriptome, proteome, and immunoproteome available to researchers so that new vaccine antigens may be discovered. Furthermore, advances in our understanding of many other characteristic features of microbes, such as epidemiology, evolution, virulence gene identification, interaction of pathogen with host, intra- and interspecies complexity, and physiology, have also helped further improve vaccine development. Epidemiological and evolutionary studies examine the microevolution of strains and the identification of newly emerging clones, using various molecular biology techniques such as multilocus sequence typing (MLST). To design a universal vaccine capable of targeting all pathogenic strains, molecular epidemiological studies consider trend changes in the population, clustering patterns (regional and worldwide), and antigenic variation among isolates. Furthermore, molecular epidemiology research can support the evaluation of a vaccines effectiveness, the incidence of vaccine escape variations, or the emergence of novel harmful variants as a result of intense selection pressure. Virulence gene identification was also made possible by complete genome sequencing of pathogenic strains. The process helped identify genetic patterns associated with disease virulence as well as the genetic elements that support immunity or a positive vaccine response. The data will help develop vaccines that have higher efficacy and specific immune responses. Recently, vaccine research and development programs are typically driven by one of these approaches or a combination of them, with the choice of technique being greatly influenced by the traits of the target pathogen. For example, the conventional RV strategy is sufficient for low-antigenic and non-cultured species, while species that have greater antigen diversity and variability may require a refined RV approach. Other approaches involving transcriptomics (RNA sequencing) and proteomics reduce the screening time of potential candidates, enabling rapid selection and evaluation of antigens.

Major barriers associated with conventional vaccine design approaches can be overcome using genomic approaches. Understanding how immunity develops is the first barrier. A possible solution may necessitate a systems biology approach involving the identification of non-humoral protection, understanding effectors and vaccines that induce cellular immunity, and clinical assessment strategies. The second barrier is host variability (age, sex, ethnic group, genetic diversity). A genomics-based solution to this includes designing population-based vaccines and carrying out diagnostic tests to predict population wide vaccine responses. The third barrier is pathogen variability (multitude of strains, varying stages of infection, different pathogen–host interactions, strain evolution). Multivalent vaccines that elicit strong immune responses in all strains (pan-genomics) and lead to the production of several neutralizing antibodies have been designed. The fourth barrier concerns vaccine safety. Vaccines may have side effects, involving autoimmune responses. Individuals may also refuse to get vaccinated owing to such concerns. To minimize safety-related issues, novel subunit and protein adjuvants may be incorporated into the design of vaccines to enhance specific immunogenicity and grant durable protection. Lastly, non-heritable factors (eating habits, alcoholism, smoking, living environment/geography, etc.) that affect vaccine efficacy are also major barriers. Genomics approaches partially resolve these issues as vaccines can be specifically designed for naïve and exposed individuals. Vitamin supplements that can boost immunity are also available.

In short, barriers associated with conventional methods were largely resolved by the application of genomics technology, which greatly aids the development of innovative vaccines. However, arduous validation and the need for animal testing and clinical trials are still major technical issues that delay vaccine production. Another issue is the limited availability of online databases containing epitope data. To enhance the reliability of the genomics approach, structural biology data, immunogenicity data, and in silico B cell/T cell epitope data are critically required. The rapid evolution and advancement of the genomics approach brought on by structural and synthetic genomics is beyond expectations. Synthetic and structural genomics are crucial tools for future vaccine development as they can address MDR issues involved in the rise in intracellular bacterial pathogens. We are hopeful that further improvements in genomic approaches will lead to advances in the field, ultimately leading to improvements in existing vaccines and the discovery of new candidates.

## Figures and Tables

**Figure 1 pharmaceutics-15-00152-f001:**
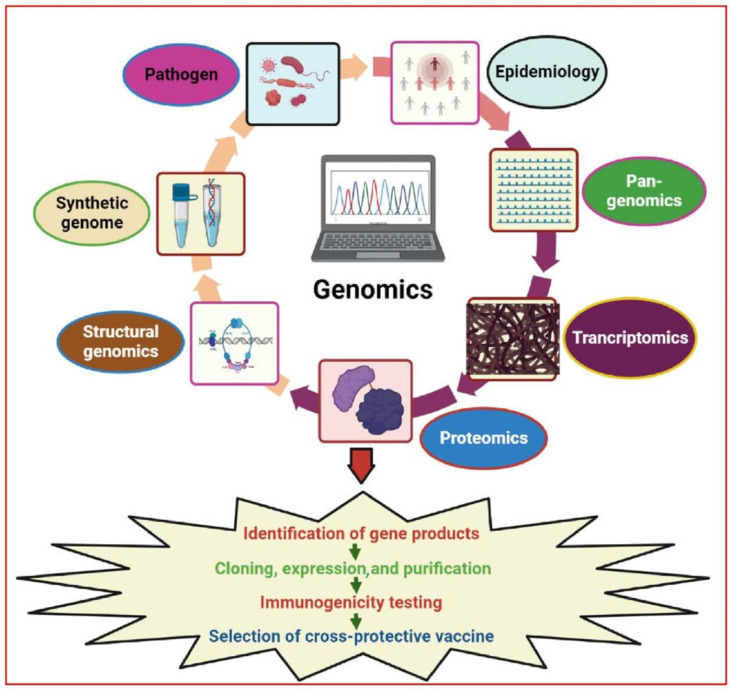
Illustration of major genomics strategies used for vaccine research and development. These strategies provide a more extensive selection of promising antigenic targets as compared to traditional vaccine development. Various technologies such as reverse vaccinology, comparative genomics, pan-genomics, functional genomics, structural vaccinology, and synthetic genomics were used for the identification of novel antigen targets. The figure was generated using BioRender software (biorender.com (accessed on 12 December 2022) and published under license agreement number (VV24R6FDFT).

**Figure 2 pharmaceutics-15-00152-f002:**
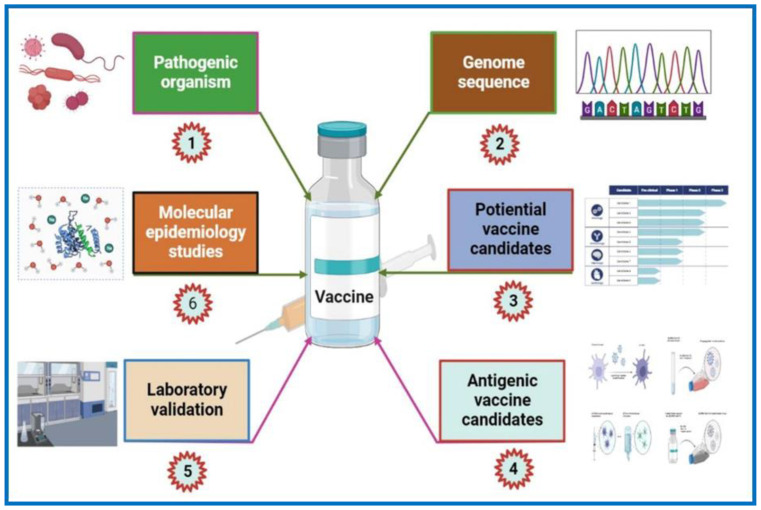
Flow chart of the reverse vaccinology approach for vaccine development. The RV approach begins with the genomics of pathogenic microorganisms using computational analysis based on the algorithms obtained from experimental biological data. Surface-associated and secretory proteins (SASPs) and virulence factors are examples of potential vaccine candidates. These are then subsequently examined to find protein candidates with B cell and T cell antigenic epitopes. The identified proteins are then PCR amplified and expressed in a vector. Further validation of these proteins includes purification followed by testing via animal models for immunogenicity. Furthermore, resulting from various experimental testing (FACS, Western blotting, serum bactericidal activity, etc.) on the animal model sera, a potential protein candidate provoking bactericidal antibodies is picked. The selected candidate undergoes a pre-clinical stage, followed by molecular epidemiological studies, and then enters into clinical trials. If the candidate shows promising results, then it undergoes the production process. The figure was generated using BioRender software (biorender.com (accessed on 12 December 2022) and published under license agreement number (OO24R9PBMM).

**Figure 3 pharmaceutics-15-00152-f003:**
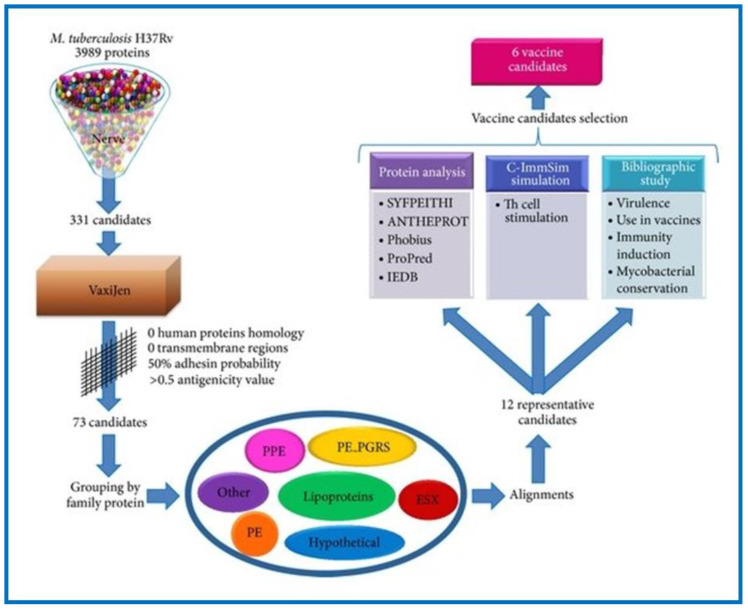
This flow diagram represents the workflow of the study conducted by Monterrubio-López and Ribas-Aparicio for the identification of promising novel vaccine candidates against *M*. *tuberculosis* through reverse vaccinology. Reprinted from Monterrubio-López and Ribas-Aparicio, 2015 [106].

**Table 3 pharmaceutics-15-00152-t003:** Bioinformatics tools available for data mining and vaccine candidate prediction. This information was collected from various sources [71,73,74,75,76,77].

Tool Category	Bioinformatics Tool	Description/Web Site
**General tools**	Basic Local Alignment Search Tool (BLAST)	Sequence similarity database http://www.ncbi.nlm.nih.gov/BLAST/ (accessed on 2 September 2022)
	Protein Subcellular Localization Predictor Tool for bacteria (PSORTb)	Bacterial protein subcellular localization (SCL) predictor http://www.psort.org/psortb/ (accessed on 2 September 2022)
	Signal peptides (SignalP)	Prediction of signal peptides http://www.cbs.dtu.dk/services/SignalP/ (accessed on 2 September 2022)
	Expert Protein Analysis System (EXPASY)	Translate nucleotide to protein http://www.expasy.org/ (accessed on 2 September 2022)
	Protein Data Bank (PDB)	Structure of protein http://www.pdb.org/pdb/home/home.do (accessed on 2 September 2022)
	Protein Variability Server (PVS)	Sequence variability calculator http://imed.med.ucm.es/PVS/ (accessed on 2 September 2022)
	Allergenic Prediction (AlgPred)	Predicting allergenic proteins and allergenic regions in a protein http://www.imtech.res.in/raghava/algpred/ (accessed on 2 September 2022)
**T cell epitope prediction**	Epitopes Major Histocompatibility Complex (EPIMHC)	MHC-binding peptides and T cell epitopes http://bio.dfci.harvard.edu/epimhc/ (accessed on 2 September 2022)
	Cytotoxic T lymphocytes Predication (CTL PRED)	Direct approach for CTL epitope prediction essential for designing subunit vaccines http://www.imtech.res.in/raghava/ctlpred/ (accessed on 2 September 2022)
	EpiVax	Epimatrix algorithm http://www.epivax.com/ (accessed on 2 September 2022)
	Kernel-based Inter-allele peptide binding prediction SyStem (KISS)	SVM-based method KISS http://cbio.ensmp.fr/kiss/ (accessed on 2 September 2022)
	Protein Predication (ProPred)	Web-based graphical prediction tool for MHC class II binding sites in antigenic protein sequences http://www.imtech.res.in/raghava/propred/ (accessed on 2 September 2022)
	IMTECH	Quantitative matrix approach http://www.imtech.res.in/raghava/mhc (accessed on 2 September 2022)
	Promiscuous EPitope-based VACcine (PEPVAC)	Prediction of promiscuous epitopes http://immunax.dfci.harvard.edu/PEPVAC/ (accessed on 2 September 2022)
	RANKPEP	Predicts peptide binders to MHCI and MHCII http://bio.dfci.harvard.edu/Tools/rankpep.html (accessed on 2 September 2022)
	SYFPEITHI	Database for searching and T cell epitope prediction http://www.syfpeithi.de/ (accessed on 2 September 2022)
	Immune Epitope Database (IEDB)	Identification of novel B and T cell epitopes in proteins of interest http://www.immuneepitope.org/ (accessed on 2 September 2022)
	NetCTL 1.2 Server	Verified Prediction of CTL epitopes in protein sequences http://www.cbs.dtu.dk/services/NetCTL/ (accessed on 2 September 2022)
	Major Histocompatibility Complex Predication (MHCPred)	A quantitative T cell epitope prediction server http://www.ddg-pharmfac.net/mhcpred/MHCPred/ (accessed on 2 September 2022)
	NetMHC 3.0	Predicts peptide binding to variety of different HLA alleles http://www.cbs.dtu.dk/services/NetMHC/ (accessed on 2 September 2022)
	Support Vector machine-based method form MHC (SVMHC)	Prediction of MHC-binding peptides https://abi.inf.uni-tuebingen.de/Services/SVMHC (accessed on 2 September 2022)
	EpiJen	Multistep algorithm for T cell epitope prediction http://www.ddg-pharmfac.net/epijen/EpiJen/ (accessed on 2 September 2022)
	Major Histocompatibility Complex Binding and Non-binding peptides (MHCBN)	Information about MHC binders, MHC non-binder, TAP binders, TAP non-binders, and T cell epitopes http://www.imtech.res.in/raghava/mhcbn/ (accessed on 2 September 2022)
**B cell epitope tools**	B-cell epitopes predication 2.0 (BepiPred 2.0)	Continuous B cell epitope identification http://www.cbs.dtu.dk/services/BepiPred/ (accessed on 2 September 2022)
	B-cell predication (Bcepred)	Continuous B cell epitope identification http://crdd.osdd.net/raghava/bcepred/ (accessed on 2 September 2022)
	B-cell Epitope prediction using Support vector machine Tool (BEST)	Continuous B cell epitope identification http://biomine.cs.vcu.edu/datasets/BEST/ (accessed on 2 September 2022)
	Continuous B-cell Epitopes Pro *(COBEpro)*	Predictions on short peptide fragments http://scratch.proteomics.ics.uci.edu/ (accessed on 2 September 2022)
	Conformational B-cell Epitope predication (CBTOPE)	Prediction of conformational B cell epitope http://crdd.osdd.net/raghava/cbtope/submit.php (accessed on 2 September 2022)
	DiscoTope 2.0	Discontinuous B cell epitopes prediction http://www.cbs.dtu.dk/services/DiscoTope/ (accessed on 2 September 2022)
	Peptide epitope (Pepitope)	Continuous and discontinuous B cell epitopes prediction http://pepitope.tau.ac.il/ (accessed on 2 September 2022)
	Epitopia	Continuous and discontinuous B cell epitope prediction http://epitopia.tau.ac.il/ (accessed on 2 September 2022)
	DiscoTope 1.2 Server	Uses 3D structure of protein to forecast discontinuous B cell epitopes http://www.cbs.dtu.dk/services/DiscoTope/ (accessed on 2 September 2022)
	Linear B-cell epitopes (LBtope)	Accurate method for linear B cell epitopes (https://webs.iiitd.edu.in/raghava/lbtope/ (accessed on 2 September 2022)
	Ellipsoid and Protrusion (ElliPro)	Method used for continuous and discontinuous B cell epitopes prediction http://www.pepitope.tau.ac.il/ (accessed on 2 September 2022)
	Artificial neural network-based B-cell epitope prediction (ABCpred)	Predicting linear B cell epitopes (https://webs.iiitd.edu.in/raghava/abcpred/ (accessed on 2 September 2022)
	Proteasomal cleavage (Pcleavage)	Proteasomal cleavage site identification (https://webs.iiitd.edu.in/raghava/pcleavage/ (accessed on 2 September 2022)
	A database for B-Cells Epitopes (BCIPEP)	B cell epitope validation http://www.imtech.res.in/raghava/bcipep/index.html (accessed on 2 September 2022)

**Table 4 pharmaceutics-15-00152-t004:** Post genomics techniques for vaccines against intracellular bacterial infections, along with the current state of each vaccine’s development. Data were collected from Bambini and Rappuoli (2009) [93].

Bacteria	Disease	Techniques	Vaccine Status
*C. pneumoniae*	Pneumonia, meningitis, middle ear infections	Reverse vaccinology Proteomics Pan-genomic	Discovery/pre-clinical
*B. anthracis*	Anthrax	Reverse vaccinology CGH microarray Microarray Proteomics Immunoproteomics Antigen prioritization	Discovery/pre-clinical
*R. prowazekii*	Epidemic typhus, also called louse-borne typhus	Reverse vaccinology Proteomics Pan-genomic	Discovery/pre-clinical
*M. tuberculosis*	Tuberculosis	Reverse vaccinology Structural genomics/vaccinology Synthetic genomics Antigen prioritization Pan-genomic	Discovery/pre-clinical
*H. pylori*	Ulcer, atrophic gastritis, adenocarcinoma, lymphoma	Reverse vaccinology Immunoproteomics Pan-genomic	Discovery/pre-clinical
*P. gingivalis*	Periodontitis	Reverse vaccinology	Discovery/pre-clinical
*N.* *meningitidis serogroup B*	Bacterial meningitis and septicemia	Reverse vaccinology Microarray Proteomics	Phase II clinical trials
*Brucella*	Brucellosis	Proteomics	Discovery/pre-clinical
*L. monocytogenes*	Invasive foodborne infections.	Proteomics	Phase 1 clinical trials

**Table 5 pharmaceutics-15-00152-t005:** Summary of antigens or developed vaccines identified by comparative genomics against various bacterial pathogens. Source data were collected from Schubert and Christodoulides, 2013 [140].

Bacteria	Developed Vaccine/Antigen Identified by Pan-Genomics	References
*N. meningitidis*	4CMenB (Bexsero^®^) vaccine	[141]
*C. pneumoniae*	LcrE antigen	[102]
*Brucella*	RV web-based vaccine design program (VaxiJen), that identified O-sialoglycoprotein endopeptidase; 32 OMPs (Omp2b, Omp25, Omp31-1, TonB, adhesins, and adhesin-like proteins (FlgE and FlgK)	[142]
*S. agalactiae*	GBS322 (SAG0032, Sip protein), GBS67 (SAG1408), GBS80	[95]
	(SAG0645), GBS104 (SAG0649) proteins	[143]
	HMW pilus-based vaccine	[144]
*S*. *pneumoniae*	Sp36, Sp46, Sp91, Sp101, and Sp128/130 protective antigens (cell wall anchor)	[89]
	RrgB321 (fusion protein of three RrgB variants)	[145]
*S. pyogenes* (*GAS*)	Cpa, MI_128, and MI_130 (Recombinant pilus protein)	[96]
	Protective antigen Spy0416	[146]
*L. pneumophila*	Q5ZVG4, Q5ZRZ1, Q5ZWE6, Q5ZT09, and Q5ZUZ8	[134]
*M.* *tuberculosis*	PE/PPE, plcC	[124]
	ESAT-6 antigen and diacylglycerol acyltransferase	[129]

## Data Availability

Not applicable.

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
