# Peer review of "Recent Advances in Genomics-Based Approaches for the Development of Intracellular Bacterial Pathogen Vaccines"

_pharmaceutics, 2022, doi:10.3390/pharmaceutics15010152_

Round 1
Reviewer 1 Report
Khan et al. provide a review on techniques used in vaccine development for intracellular bacterial pathogens. There are many grammatical errors (too many to note) throughout the entire work that make sections unclear and difficult to follow. Professional editing services are strongly recommended. In general, there is not enough information described. Frequently, I found myself wondering what became of the research described (clinical trials, approved vaccine, etc.), and wanting/needing more detailed information to understand key concepts. In addition, a critical review of the described literature is lacking and there is little to no attempt to synthesize new information. I have the following comments and suggestions.
Throughout the entire document, bacterial names (genus and species) must be italicized. The first time it appears, the full genus and species are written out and for the following mentions, they are abbreviated (i.e. S. aureus).
Please confirm that the first sentence in the abstract is actually a fact.
Page 2: Paragraph 1, last two sentences. These are weak concluding sentences. The life-styles of most bacteria are now well known, as is much about their virulence. These sentences can be deleted.
Page 2, Paragraph 2: In the list of facultative intracellular bacteria, Legionella pneumophila is listed twice. The abbreviation for species is spp, not ssp. Second to last sentence: Do you really mean fetal diseases? All of those can cause infections in adults. Why are you discussing the incubation times of the bacteria? How is it relevant?
Page 2, Paragraph 3: Please add more detail about the conventional methods for vaccine development and discuss some examples that have come out of these methods.
Page 2, Paragraph 4: What is meant by the “method requiring several years?” Please expand up this statement. Is that for development or production?
Page 3: Please add a detailed section on “Traditional Vaccine Development.” You need this foundation to understand why genomics techniques provide an advantage.
Table 1: Please fix chart so that the end of a word is on the same line as the word. Italicize species. Why is the extracellular bacteria S. aureus included in this table (mentioned in introduction)? M. tuberculosis is not really Gram-positive. It is acid fast. Rickettsia and Chlamydia are usually referred to as Gram-indeterminant, because they stain very weakly with Gram stain.
Table 2: This does not contain all intracellular bacteria, as the heading suggests. Would it be possible to add references and additional information about where the vaccines are approved? If the names are provided for two licensed ones, it should be given for all. What is 85A antigen? The “bacterial vectors” section is unclear as is the plasmid DNA. Maybe remove the information in parentheses and include a better description as part of a table legend.
Page 5: What is meant by “target 389?” It is mentioned that genome-based approaches will identify 10-100 more candidates. How? Elaborate and give specific examples. Where Table 3 is discussed. These resources should be thoroughly discussed. Which are the most commonly used or are the best and why? Are any of these new approaches?
Figure 1 looks odd with two sections blank on the wheel. Can you remake with only 5 sections?
Table 2: Fix so all websites are on one line.
Page 7: The statement about infectious diseases being the biggest cause of death globally needs references. Please add details of how non-cultivatable organisms can be studied using genomic techniques. Describe the advantages of using genomic techniques in more detail. Last sentence, change “short cut” to “foundation.”
Page 8, paragraph 1: Describe how the RV technique is performed in more detail.
Page 8, paragraph 2: I don’t understand how capsular polysaccharides could be problematic. These polysaccharides are poor immunogens, which is why the capsule is an effective strategy of bacteria to avoid the immune system. You mention that 30 proteins resulted in antibody production that could eliminate the bacteria in vitro. How was this determined?
Page 8, paragraph 3: The first sentence does not make sense. Please rewrite. Give more details about the status of each potential vaccine (in Table 4). How far has each gone? Which phase of clinical trials? A reference is needed when discussing GAS. Give details on the molecular mimicry, M protein and rheumatic fever. Protein-based vaccines: discuss in more detail and provide a critical review. Are they effective or promising? What are the shortcomings or disadvantages?
Page 8, paragraph 4: First sentence needs a reference. I am not sure that is a fact. It should be mentioned that the BCG vaccine is not universally approved. Mention where it is used.
Page 9, paragraph 1: How do you know EsxL, PE26, etc., will be more effective as vaccine targets? This statement is too strong. Also, at the end, were any of these approved or did they get to phase III? Add details. Strep. pyogenes should be written as S. pyogenes. Change this is the next paragraph too.
Page 9 paragraph 2: What is meant by “…with each new genome displaying 18% new genes?” Is this refereeing to strains of bacteria and that they vary by 18%? Was the GAS vaccine ever in clinical trials or used?
Table 4: Which phase of clinical trials is the listeria vaccine in? Make sure all letters in a word are on the same line.
Page 11, paragraph 1: The opening paragraph is vague and not well related to vaccines. Please add some more detail.
Page 11, paragraph 2: This research was performed in 2005. So, what became of it? Did it lead to the production of a used vaccine? Please make sure to include this type of information.
Page 11, paragraph 3: It is not relevant to vaccine development that pan-genomics can be used to understand pathogen history or support diagnoses.
Page 12, paragraph 1: This information is not relevant.
Page 12, paragraph 2: What is PPE19 and plc? Any time a new gene/protein is discussed, it must be described. What was the result of this work? Is it headed to clinical trial, or do you think it should?
Page 12, paragraph 3: Instead of “indigestive problems,” use gastroenteritis. What is E-CGFs? Add more details on the studies.
Page 12, paragraph 4: What specifically do you mean by “constitutes 90% of infections in humans?” This statement is not true as written. What is subtractive proteomics? Add more details. So, is the proposed Legionella vaccine promising or heading to clinical trials? Rickettsia rickettsii causes RMSF. When it says genomes, does that mean strains, clinical isolates, or something else? What are the next steps with this potential vaccine?
Page 12, paragraph 5: The second sentence should be expanded upon and explained better. The last sentence is not relevant to the review and should be removed.
Page 12, paragraph 6: What became of the study? Is it in clinical trials, or approved for use?
Page 13, “antigens prioritization” section: Explain “tic disorder.” Which GAS antigen are your referring to? Explain the GAS microarray process in more detail. I do not understand this sentence “…and the downfall with this process is the expression of all proteins which further deteriorate the proper folding.” Please rewrite for clarity. At the end of the paragraph, was the technique effective for any of those bacteria?
Page 14, paragraph 1: For the sake of a non-expert reader, please describe passive immunization.
Page 14, paragraph 3: Complete microbial genome sequences are not available because of microarray technology. They are available because of whole genome sequencing.
Page 14, paragraph 4: What is every MenB gene? That is shorthand for a serotype. It should be listed as every N. meningitidis gene. Later in the paragraph, DEGs are mentioned. Which ones are of interest? What became of the research described in the last sentence?
Page 14, paragraph 5: In strain UAMS-1, which exotoxins? How did the gene expression data shed light on pathogenesis emphasize the value for vaccine development? Later, when discussing M. tuberculosis, how did transcriptomics shed light on the immune system response? It is mentioned at the end that the benefits of transcriptomics increase with increased technology. What benefits? In addition, microarray technology is an older technology. I found it very surprising that the potential of advanced transcriptomics using Next-gen sequencing (i.e. RNA-seq) was not discussed at all.
Page 15, paragraph 1: When talking about the proteomic approach, the protein mixture is not “broken down first into its component parts.” 2D gel electrophoresis or chromatography is used to reduce complexity of the samples by separating proteins. The mass spectrometry section should be described in more detail. For example, a non-expert will not understand what a peptide-mass fingerprint is. Explain the “surfome” technique in more detail. Last sentence-explain this in more detail.
Page 15, paragraph 2: The Chlamydia scenario lists details with no context, making understanding this section very challenging. Please make sure to explain everything that is discussed.
Page 15, paragraph 3: Where is the reference that Salmonella is the most investigated bacteria for proteomic analysis? Are you referring to just for vaccine development? E. coli is likely the most thoroughly investigated bacteria using proteomics approaches.
Page 16, paragraph 1: What are ssaV, sseF and SrfN? When discussing genes, they should be italicized. Last sentence, what came of that research? Were vaccines developed and successfully used?
Page 16, paragraph 2-3: A more detailed explanation of SV should be included. These two paragraphs can be combined into 1.
Page 16, paragraph 4 onto page 17. There are approved MenB vaccines available and they should be discussed. Which of these studies led to vaccine development?
Page 17, paragraph 2: it is mentioned that ‘the study suggests a vaccine candidate…” Which one? Last sentence-how did the sort out the structure as a useful therapeutic target.
Page 17, paragraph 3: This paragraph needs to be completely rewritten. It is over speculation and non-statements. A few concluding statements about potential usefulness or limits of this approach would be better here.
Page 17, paragraph 6: First sentence: remove “pathogens.” A reference is needed for this statement.
Page 18, 1st sentence: What is TDR? Define MDR and XDR as well. Add more details.
Page 18, paragraph 1: The reference to the COVID pandemic is not relevant.
Page 18, paragraph 2: How does advancement in understanding of epidemiology, evolution, and virulence genes aid in vaccine development? Explain more. What is meant by “lower down the screening?”
Page 18, paragraph 3: What shortcomings of conventional methods? This is never addressed. How did genomic technology address these? How many vaccines were developed using this new technology compared to the old one? Validation, animal models and clinical trials are not issues that researchers are facing, they are needed for safety and efficacy before widespread use. This section should be a critical evaluation of the techniques for vaccine development (pros and cons, and what sort of things are need to advance the field).

Author Response
Reviewer 1
Khan et al. provide a review on techniques used in vaccine development for intracellular bacterial
pathogens. There are many grammatical errors (too many to note) throughout the entire work that make sections unclear and difficult to follow. Professional editing services are strongly recommended. In general, there is not enough information described. Frequently, I found myself wondering what became of the research described (clinical trials, approved vaccine, etc.), and wanting/needing more detailed information to understand key concepts. In addition, a critical review of the described literature is lacking and there is little to no attempt to synthesize new information. I have the following comments and suggestions.
Throughout the entire document, bacterial names (genus and species) must be italicized. The first time it appears, the full genus and species are written out and for the following mentions, they are abbreviated (i.e. S. aureus).
Author’s Response
We thank the reviewer for the comment and suggestion. The names of the bacteria have now been corrected according to your valuable suggestion.
Please confirm that the first sentence in the abstract is actually a fact.
Author’s Response
We thank the reviewer for the comment and suggestion. The first sentence in the abstract has been revised.
Page 2: Paragraph 1, last two sentences. These are weak concluding sentences. The life-styles of most bacteria are now well known, as is much about their virulence. These sentences can be deleted.
Author’s Response
We thank the reviewer for the comment and suggestion. Some minor discrepancies or inconsistencies are present in bacterial classifications based on the bacterial lifestyle. A paragraph has been included in the introduction regarding the discrepancies in the classifications. Information regarding some methods to further classify the bacteria has also been included.
Page 2, Paragraph 2: In the list of facultative intracellular bacteria, Legionella pneumophila is listed twice. The abbreviation for species is spp, not ssp. Second to last sentence: Do you really mean fetal diseases? All of those can cause infections in adults.
Author’s Response
We thank the reviewer for the comment and highlighting some of our mistakes. The listing error has been corrected. The species abbreviations have also been corrected. “Do you really mean fetal diseases” - This was a spelling error. The correct word is “Fatal”.
Why are you discussing the incubation times of the bacteria? How is it relevant?
We discuss the various characteristic features of intracellular bacteria such as Intracellular lifestyles, Bacterial factors, Host Cells localization and Incubation Time. The inclusion of bacterial incubation time is relevant as it is stated in the literature that effective vaccine development for pathogens depends on the incubation times.
Page 2, Paragraph 3: Please add more detail about the conventional methods for vaccine development and discuss some examples that have come out of these methods.
Author’s Response
We thank the reviewer for the comment and suggestion. We have now included more details about the conventional methods for vaccine development and have provided examples.
Page 2, Paragraph 4: What is meant by the “method requiring several years?” Please expand up this statement. Is that for development or production?
Author’s Response
We thank the reviewer for the comment. The statement has been modified in the revised manuscript for clarity. A longer duration is required for the production process.
Page 3: Please add a detailed section on “Traditional Vaccine Development.” You need this foundation to understand why genomics techniques provide an advantage.
Author’s Response
We thank the reviewer for the comment. A Traditional Vaccine Development section has been included in the revised manuscript.
Table 1: Please fix chart so that the end of a word is on the same line as the word. Italicize species. Why is the extracellular bacteria S. aureus included in this table (mentioned in introduction)? M. tuberculosis is not really Gram-positive. It is acid fast. Rickettsia and Chlamydia are usually referred to as Gram-indeterminant, because they stain very weakly with Gram stain.
Author’s Response
We thank the reviewer for the comment. Table 1 has been modified so that all of the words end on the same line. The names of all species have been italicized. In the introduction section S. aureus was mentioned by mistake. Many studies have reported the intracellular nature of S. aureus. References are provided here.
Bravo-Santano, N., Ellis, J. K., Mateos, L. M., Calle, Y., Keun, H. C., Behrends, V., & Letek, M. (2018). Intracellular Staphylococcus aureus modulates host central carbon metabolism to activate autophagy. Msphere, 3(4), e00374-18.
Neumann, Y., Bruns, S. A., Rohde, M., Prajsnar, T. K., Foster, S. J., & Schmitz, I. (2016). Intracellular Staphylococcus aureus eludes selective autophagy by activating a host cell kinase. Autophagy, 12(11), 2069-2084.
Rollin, G., Tan, X., Tros, F., Dupuis, M., Nassif, X., Charbit, A., & Coureuil, M. (2017). Intracellular survival of Staphylococcus aureus in endothelial cells: a matter of growth or persistence. Frontiers in microbiology, 8, 1354.
The gram negative and positive nature of the bacteria listed in table 1 were updated according to the new information.
Table 2: This does not contain all intracellular bacteria, as the heading suggests. Would it be possible to add references and additional information about where the vaccines are approved? If the names are provided for two licensed ones, it should be given for all. What is 85A antigen? The “bacterial vectors” section is unclear as is the plasmid DNA. Maybe remove the information in parentheses and include a better description as part of a table legend.
Author’s Response
We thank the reviewer for the comment and suggestion. We modified the title of the table as it contains both intracellular and extracellular bacteria. The references are now included in the revised manuscript and more vaccine types have been added to the table. The names of all the licensed vaccines have now been provided in the revised manuscript, with references. Ag85A, induces strong T-cell proliferation and IFN-γ production in individuals infected with M. tuberculosis and in those vaccinated with BCG. Ag85A is regarded as a promising TB vaccine candidate.
Page 5: What is meant by “target 389?” It is mentioned that genome-based approaches will identify 10-100 more candidates. How? Elaborate and give specific examples. Where Table 3 is discussed. These resources should be thoroughly discussed. Which are the most commonly used or are the best and why? Are any of these new approaches?
Author’s Response
We thank the reviewer for the comment and suggestions. We apologize for our typing error. Target 389 was corrected to “389 (0.2%) targeted gene sequencing”, a sequencing project in the online genome database (https://gold.jgi.doe.gov/index).
We have included the following sentence for further clarification and have provided a reference.
“Advancements in long reading sequencing technologies will facilitate the systematic assembly of diploid genomes, revolutionizing genomics by presenting the entire range of human genetic variation, covering some unaccounted heritability and uncovering novel disease processes.”
Seib, K. L., Dougan, G., Rappuoli, R. (2009). The key role of genomics in modern vaccine and drug design for emerging infectious diseases. PLoS Genet. 5, e1000612.
With respects to Table 3, it has been cited in the text and the function of each tool has now been explained in the revised manuscript. The basic role of these tools and resources were explained in table 3 as these were the primary data acquisition and analysis tools in vaccine development. Table 3 was also cited in the RV section for data mining.
Figure 1 looks odd with two sections blank on the wheel. Can you remake with only 5 sections?
Author’s Response
We thank the reviewer for the comment and suggestion. Figure 1 has been rectified in the revised manuscript according to your valuable suggestion.
Table 3: Fix so all websites are on one line.
Author’s Response
We thank the reviewer for the comment and suggestion. Table 3 was reformatted in the revised manuscript.
Page 7: The statement about infectious diseases being the biggest cause of death globally needs
references. Please add details of how non-cultivatable organisms can be studied using genomic
techniques. Describe the advantages of using genomic techniques in more detail. Last sentence, change “short cut” to “foundation.”
Author’s Response
We thank the reviewer for the comment and suggestion. A reference has been added. More details regarding non-culturable organisms and their genomics have been added in the revised manuscript. Additional advantages of the different genomic techniques have also been included in the revised manuscript. The word “short cut” has been replaced with “foundation”.
Page 8, paragraph 1: Describe how the RV technique is performed in more detail.
Author’s Response
We thank the reviewer for the comment and suggestion. A graphical illustration of RV has now been included in the revised manuscript.
Page 8, paragraph 2: I don’t understand how capsular polysaccharides could be problematic. These
polysaccharides are poor immunogens, which is why the capsule is an effective strategy of bacteria to avoid the immune system.
Author’s Response
We thank the reviewer for the comment and suggestion. The relevant information has now been included in the revised manuscript.
“Based on the chemical composition of polysaccharide capsules, N. meningitidis was identified as having 13 serogroups. Only five serogroups (A, B, C, Y, and W135) have been linked to meningococcal meningitis and sepsis. Capsular polysaccharides have been used in vaccines against serogroups A, C, Y, and W135 in both adults and infants. Interestingly, it was found that when the bacterial polysaccharide was conjugated with a carrier protein, a strong T cell-dependent immune response was evoked following vaccination that provided long-term protection. However, such an immune response was not observed serogroup B bacteria. Initial attempts to develop a Meningococcal B vaccine by conventional methods were unsuccessful. Although there were several reasons for this failure, two primary reasons were apparent. First, meningococcal B capsular polysaccharides (CPS) (a polymer of α (2‑8) linked N‑acetylneuraminic acid) had a high similarity to components of human tissues, which resulted in poor immunogenicity in humans, often stimulating autoimmune responses. Secondly, as protein-based vaccines are highly antigen-specific, they only provide defense against a very small number of strains (Romero and Outschoorn, 1994).”
. You mention that 30 proteins resulted in antibody production that could eliminate the bacteria in vitro. How was this determined?
Details of the RV approach protocol with respects to MenB have now been included in the revised manuscript.
“Out of these 600 genes, 350 were expressed successfully in E. coli, and were subsequently used to examine immunogenicity in mice (Pizza et al., 2000). Enzyme linked immune sorbent assay (ELISA), fluorescence-activated cell sorter (FACS), and immunoblot analysis were performed using the immune serum of the mice to examine surface-exposed localization (Pizza et al., 2000). These experiments identified 90 previously unknown surface localized proteins. Bactericidal assays and/or passive protection in infant rat assays were subsequently carried out to identify that 30 out of the 90 novel proteins triggered the production of antibodies which could eradicate the bacteria in vitro.”
Page 8, paragraph 3: The first sentence does not make sense. Please rewrite. Give more details about the status of each potential vaccine (in Table 4). How far has each gone? Which phase of clinical trials? A reference is needed when discussing GAS. Give details on the molecular mimicry, M protein and rheumatic fever. Protein-based vaccines: discuss in more detail and provide a critical review. Are they effective or promising? What are the shortcomings or disadvantages?
Author’s Response
We thank the reviewer for the comment and suggestion. The sentence has been modified in the revised manuscript. Most of the vaccines are in discovery phase, while some are in the clinical trial phase I and II as listed in table 4. A statement has also been included in the text. A reference regarding GAS has also been include in the revised manuscript. Some details on molecular mimicry, M proteins and rheumatic fever have also been included in the revised manuscript to clarify the basic concept. Discussion regarding proteins-based vaccines has also been breifly included along with the respective disadvantages. Complete details have neem included under the section “Functional Genomics” section sub heading “Proteomics; A genomic complement for vaccine development”.
Page 8, paragraph 4: First sentence needs a reference. I am not sure that is a fact. It should be mentioned that the BCG vaccine is not universally approved. Mention where it is used.
Author’s Response
We thank the reviewer for the comment and suggestion. A reference has been provided and the sentence has been modified according to the updated information on WHO website. The statement is included in the revised manuscript and information regarding BCG usage has also been included. Furthermore, the web page reference regarding the usage of BCG in every country is also mentioned.
Page 9, paragraph 1: How do you know EsxL, PE26, etc., will be more effective as vaccine targets? This statement is too strong. Also, at the end, were any of these approved or did they get to phase III? Add details. Strep. pyogenes should be written as S. pyogenes. Change this is the next paragraph too.
Author’s Response
We thank the reviewer for the comment. These candidates were filtered based on the various characteristics that were examined by (Monterrubio-Lopez and Ribas-Aparicio, 2015), such as Adhesin probability (%), Antigenicity value (VaxiJen), Th1 cell number stimulated (1/3 injections), M. bovis conservation, BCG conservation, Immunologic relation, function, and extensive research on the literature. Based on these characteristics we rationalized that these 6 candidates were selected and that they will improve the design of novel TB vaccines. Details about the vaccine status have now been included in the revised manuscript with respects to these six candidates. The name of the bacteria was corrected according to your valuable suggestion.
Page 9 paragraph 2: What is meant by “…with each new genome displaying 18% new genes?” Is this refereeing to strains of bacteria and that they vary by 18%? Was the GAS vaccine ever in clinical trials or used?
Author’s Response
We thank the reviewer for the comment and suggestion. Yes, 18% diversity was found in at least one of the strains. Yes, some GAS vaccines have been used in the clinical trials. We have incorporated this information in the revised manuscript.
Table 4: Which phase of clinical trials is the listeria vaccine in? Make sure all letters in a word are on the same line.
Author’s Response
We thank the reviewer for the comment and suggestion. The phase information of listeria vaccine has been included in the manuscript and the table has been reformatted.
Page 11, paragraph 1: The opening paragraph is vague and not well related to vaccines. Please add some more detail.
Author’s Response
We thank the reviewer for the comment and suggestion. More information related to pan genomics or comparative genomics has been included in the revised manuscript.
Page 11, paragraph 2: This research was performed in 2005. So, what became of it? Did it lead to the production of a used vaccine? Please make sure to include this type of information.
Author’s Response
We thank the reviewer for the comment and suggestion. The requested information has now been included.
Page 11, paragraph 3: It is not relevant to vaccine development that pan-genomics can be used to
understand pathogen history or support diagnoses.
Author’s Response
We thank the reviewer for the comment and suggestion. The irrelevant information has been removed from the revised manuscript.
Page 12, paragraph 1: This information is not relevant.
Author’s Response
We thank the reviewer for the comment and suggestion. The irrelevant information has been removed from the revised manuscript.
Page 12, paragraph 2: What is PPE19 and plc? Any time a new gene/protein is discussed, it must be described. What was the result of this work? Is it headed to clinical trial, or do you think it should?
Author’s Response
We thank the reviewer for the comment and suggestion. We have now described the genes in greater detail, along with their specific function. The result has also been explained and a conclusion has been drawn.
Page 12, paragraph 3: Instead of “indigestive problems,” use gastroenteritis. What is E-CGFs? Add more details on the studies.
Author’s Response
We thank the reviewer for the comment and suggestion. The term indigestive problem has been replaced with gastroenteritis. More details related to the studies have been included and the term E-CGFs has been explained in the revised manuscript.
Page 12, paragraph 4: What specifically do you mean by “constitutes 90% of infections in humans?” This statement is not true as written. What is subtractive proteomics? Add more details. So, is the proposed Legionella vaccine promising or heading to clinical trials? Rickettsia rickettsii causes RMSF. When it says genomes, does that mean strains, clinical isolates, or something else? What are the next steps with this potential vaccine?
Author’s Response
We thank the reviewer for the comment and suggestion. The statement has been rectified. References to the statement have now been included. Details regarding subtractive proteomics have also been included in the revised manuscript. Computer processing has predicated that this vaccine may be promising, further wet lab studies have been recommended. These details have been included. The sentence ‘Rickettsia rickettsii causes Rocky Mountain spotted fever (RMSF).” Has been incorporated. When we state genomes in pan genomics, we are referring to different strains of the same species. The prospects of the potential vaccine have now been explained in the revised manuscript.
Page 12, paragraph 5: The second sentence should be expanded upon and explained better. The last sentence is not relevant to the review and should be removed.
Author’s Response
We thank the reviewer for the comment and suggestion. The sentence has been explained, and the results of the study had been completely elucidated. The details have been modified to make it more relevant to the review.
Page 12, paragraph 6: What became of the study? Is it in clinical trials, or approved for use?
Author’s Response
We thank the reviewer for the comment and suggestion. The study is a predictive study for potential vaccine development. New information has been included regarding the vaccine clinical trials.
Page 13, “antigens prioritization” section: Explain “tic disorder.” Which GAS antigen are your referring to? Explain the GAS microarray process in more detail. I do not understand this sentence “…and the downfall with this process is the expression of all proteins which further deteriorate the proper folding.” Please rewrite for clarity. At the end of the paragraph, was the technique effective for any of those bacteria?
Author’s Response
We thank the reviewer for the comment and suggestion. Tourette Syndrome (TS) has been described. The GAS antigens have also been elaborated in the manuscript. The reference regarding the GAS microarray process has been explained in greater detail. The microarray system protocol is complex and outside the scope of the review. The protocol has been referenced for the benefit of the reader. The sentence was rewritten to make the statement more clear.
Page 14, paragraph 1: For the sake of a non-expert reader, please describe passive immunization.
Author’s Response
We thank the reviewer for the comment and suggestion. Passive immunity has now been explained in greater detail.
Page 14, paragraph 3: Complete microbial genome sequences are not available because of microarray technology. They are available because of whole genome sequencing.
Author’s Response
We thank the reviewer for the comment. The information has been modified in the revised manuscript.
Page 14, paragraph 4: What is every MenB gene? That is shorthand for a serotype. It should be listed as every N. meningitidis gene. Later in the paragraph, DEGs are mentioned. Which ones are of interest? What became of the research described in the last sentence?
Author’s Response
We thank the reviewer for the comment. The sentence has been modified according to your suggestion. The DEGs number was mentioned along with its importance. The results of the research were also elaborated as requested.
Page 14, paragraph 5: In strain UAMS-1, which exotoxins? How did the gene expression data shed light on pathogenesis emphasize the value for vaccine development? Later, when discussing M. tuberculosis, how did transcriptomics shed light on the immune system response? It is mentioned at the end that the benefits of transcriptomics increase with increased technology. What benefits? In addition, microarray technology is an older technology. I found it very surprising that the potential of advanced transcriptomics using Next-gen sequencing (i.e. RNA-seq) was not discussed at all.
Author’s Response
We thank the reviewer for the comment. The requested information has now been included. We have attempted to address other concerns of the reviewer where possible. RNA seq information has now been included in the revised manuscript.
Page 15, paragraph 1: When talking about the proteomic approach, the protein mixture is not “broken down first into its component parts.” 2D gel electrophoresis or chromatography is used to reduce complexity of the samples by separating proteins. The mass spectrometry section should be described in more detail. For example, a non-expert will not understand what a peptide-mass fingerprint is. Explain the “surfome” technique in more detail. Last sentence-explain this in more detail.
Author’s Response
We thank the reviewer for the comment and suggestion. Details regarding the proteomic approach have been modified to make the statements clearer. Details regarding mass spectrometry have also been included. Details regarding the Surfaceome have also been improved.
Page 15, paragraph 2: The Chlamydia scenario lists details with no context, making understanding this section very challenging. Please make sure to explain everything that is discussed.
Author’s Response
We thank the reviewer for the comment and suggestion. The section regarding Chlamydia has been explained in greater detail. Language errors have also been corrected.
Page 15, paragraph 3: Where is the reference that Salmonella is the most investigated bacteria for
proteomic analysis? Are you referring to just for vaccine development? E. coli is likely the most thoroughly investigated bacteria using proteomics approaches.
Author’s Response
We thank the reviewer for the comment and suggestion. The reference has been included in the revised manuscript.
Page 16, paragraph 1: What are ssaV, sseF and SrfN? When discussing genes, they should be italicized. Last sentence, what came of that research? Were vaccines developed and successfully used?
Author’s Response
We thank the reviewer for the comment and suggestion. These are three different strains of Salmonella. The studies were predictive, based on a proteomic approach. It is likely that the findings will lead to future vaccine development.
Page 16, paragraph 2-3: A more detailed explanation of SV should be included. These two paragraphs can be combined into 1.
Author’s Response
We thank the reviewer for the comment and suggestion. SV has been described in greater detail and both paragraphs have been merged.
Page 16, paragraph 4 onto page 17. There are approved MenB vaccines available, and they should be discussed. Which of these studies led to vaccine development?
Author’s Response
We thank the reviewer for the comment and suggestion. We have mentioned that an approved vaccine against N. meningitidis serogroups (A, C, Y, and W135) is available, however it cannot be used against MenB. The RV based studies leading to the development of a MenB vaccine have been explained in the revised manuscript in the “An in-silico Approach: Reverse Vaccinology (RV)” section and in table 4 (vaccine status).
Page 17, paragraph 2: it is mentioned that ‘the study suggests a vaccine candidate…” Which one? Last sentence-how did the sort out the structure as a useful therapeutic target.
Author’s Response
We thank the reviewer for the comment. We have amended the sentences to make the statement more clear. The requested information has now been provided in the revised manuscript.
Page 17, paragraph 3: This paragraph needs to be completely rewritten. It is over speculation and non-statements. A few concluding statements about potential usefulness or limits of this approach would be better here.
Author’s Response
We thank the reviewer for the comment. The paragraph has been modified according to your valuable suggestions.
Page 17, paragraph 6: First sentence: remove “pathogens.” A reference is needed for this statement.
Author’s Response
We thank the reviewer for the comment. The term pathogens has been removed, and a reference has been included in the revised manuscript.
Page 18, 1st sentence: What is TDR? Define MDR and XDR as well. Add more details.
Author’s Response
We thank the reviewer for the comment. The terms have now been defined, and more details have been added.
Page 18, paragraph 1: The reference to the COVID pandemic is not relevant.
Author’s Response
We thank the reviewer for the comment. The statement has been modified and the irrelevant information has been removed.
Page 18, paragraph 2: How does advancement in understanding of epidemiology, evolution, and virulence genes aid in vaccine development? Explain more. What is meant by “lower down the screening?”
Author’s Response
We thank the reviewer for the comment. The statements related to the role of epidemiology, evolution and virulence genes in vaccine development have been described in greater details. The language issues have now been resolved as follows:
“Other approaches involving transcriptomics (RNA sequencing) and proteomics reduce the screening time of potential candidates enabling rapid selection and evaluation of antigens.”
Page 18, paragraph 3: What shortcomings of conventional methods? This is never addressed. How did genomic technology address these? How many vaccines were developed using this new technology compared to the old one? Validation, animal models and clinical trials are not issues that researchers are facing, they are needed for safety and efficacy before widespread use. This section should be a critical evaluation of the techniques for vaccine development (pros and cons, and what sort of things are need to advance the field).
Author’s Response
We thank the reviewer for the comment and suggestions. The shortcomings of the conventional method have now been included in the revised manuscript. The benefits of the genomic methods have also been included in the revised manuscript under the section “Genome-based approaches”. The advantages and disadvantages of genomics have also been included in the revised manuscript.
We would like to thank the reviewer for their valuable feedback. Your suggestions have drastically improved our manuscript. We envisage the adequacy of our modifications.
Thank you!
Reviewer 2 Report
General Comments
This article by M.Khan et al., aims to review recent advances in genomic based approaches for the development of intracellular bacterial vaccines. The study summarizes bacterial diseases and vaccines availability for the same. The study provided different types of vaccines targeting intracellular bacteria. The article summarized different genome approaches which can be used in the development of different vaccines.
Overall recommendation: Recommendation is Major Revision
The study is good attempt for providing a summarized data on recent genomic approaches in vaccine development targeting intracellular bacterial pathogens, however a detailed attempt to critically appraise the approaches is missing. There are many reviews in this domain in prior art. Unique selling points of the article as compared to prior art was not evidenced and or presented. Authors should try to analyze the information and some perspective should be presented.
Comments
Following are some comments which could be used during revision.
· Abstract: Rewrite the abstract section highlighting the importance of the topic and attempts made to justify the topic. Highlight the work summarized in the article. Abstract needs rewriting with detailed specific instead of broad statements.
* Correct terminology: vaccine candidate
· Please refer and use scientific way of writing any organism scientific name as binomial nomenclature (The scientific names of species are italicized. The genus name is always capitalized and is written first; the specific epithet follows the genus name and is not capitalized).
Other Technical Comments
· Introduction, Page 2: Provide statements with appropriate reference supporting numerical data. For example, “nearly half of the infectious diseases * “This is possible” mentioned statement lack clarity.
· Include a section on challenges and importance of vaccine for disease targeted by intracellular pathogens.
· Introduction section, Paragraph 2, provides the mechanism of intracellular and extracellular pathogens which shows similar information reviewed by other articles (Anke Osterloh, MDPI Vaccines, 2022.) Please include further detailed information.
· Table 1 provides summary of intracellular bacteria and illness associated, although the similar information is summarized by other article (Table S1. Anke Osterloh, MDPI Vaccines, 2022.) Please revise the table with additional information.
· Table 3 provides bioinformatics tools available for mining of data which is a good attempt, although similar information is summarized by other article (Table S1. Anke Osterloh, MDPI Vaccines, 2022). No new tool is included in the table 3. Please revise the table with additional information such as provide any examples in vaccine development where particular tool is used or it can be used.
· Figure 1 looks adopted from Guangii Lu et al., 2020. Revise the diagram or provide reference to the legend of figure; ·Figure 2 is adopted from M.Lopez et al., 2015. Include more recent examples; Correct the sentences “pertussis toxins were developed”. Page 3· Correct sentences “Pathogen”. Page 3
Author Response
Reviewer 2
General Comments
This article by M. Khan et al., aims to review recent advances in genomic based approaches for the development of intracellular bacterial vaccines. The study summarizes bacterial diseases and vaccines availability for the same. The study provided different types of vaccines targeting intracellular bacteria. The article summarized different genome approaches which can be used in the development of different vaccines.
Author’s Response
We thank the reviewer for the positive feedback. We will consider and implement the reviewers’ valuable suggestions for further improvement.
Overall recommendation: Recommendation is Major Revision
The study is good attempt for providing a summarized data on recent genomic approaches in vaccine development targeting intracellular bacterial pathogens, however a detailed attempt to critically appraise the approaches is missing. There are many reviews in this domain in prior art. Unique selling points of the article as compared to prior art was not evidenced and or presented. Authors should try to analyze the information and some perspective should be presented.
Comments
Following are some comments which could be used during revision.
- Abstract: Rewrite the abstract section highlighting the importance of the topic and attempts made to justify the topic. Highlight the work summarized in the article. Abstract needs rewriting with detailed specific instead of broad statements.
Author’s Response
We thank the reviewer for their suggestion. We have made significant modifications to the abstract. We have improved the quality of English and have also attempted to highlight the importance of the topic.
* Correct terminology: vaccine candidate
Author’s Response
We thank the reviewer for the comment. The terminology has been corrected in accordance to your suggestion.
- Please refer and use scientific way of writing any organism scientific name as binomial nomenclature (The scientific names of species are italicized. The genus name is always capitalized and is written first; the specific epithet follows the genus name and is not capitalized).
Author’s Response
We thank the reviewer for identifying the error. The scientific names of all organism’s were checked and corrected in the revised manuscript.
Other Technical Comments
- Introduction, Page 2: Provide statements with appropriate reference supporting numerical data. For example, “nearlyhalf of the infectious diseases * “This is possible” mentioned statement lack clarity.
Author’s Response
We thank the reviewer for the comment. We have now modified the sentence to show clarity and a reference has also been provided.
“The majority of infectious diseases that affect humans are caused by bacteria”.
- Include a section on challenges and importance of vaccine for disease targeted by intracellular pathogens.
Author’s Response
We thank the reviewer for the comment and suggestion. The requested information has been include in the revised manuscript.
- Introduction section, Paragraph 2, provides the mechanism of intracellular and extracellular pathogens which shows similar information reviewed by other articles (Anke Osterloh, MDPI Vaccines, 2022.) Please include further detailed information.
Author’s Response
We thank the reviewer for the comment and suggestion. The requested information has now been include in the revised manuscript.
- Table 1 provides summary of intracellular bacteria and illness associated, although the similar information is summarized by other article (Table S1. Anke Osterloh, MDPI Vaccines, 2022.) Please revise the table with additional information.
Author’s Response
We thank the reviewer for the comment and suggestion. Table 1 has been revised according to your valuable suggestion. We have now included new information about the intracellular bacterial pathogens (intracellular lifestyles, bacterial factors) for your further consideration.
- Table 3 provides bioinformatics tools available for mining of data which is a good attempt, although similar information is summarized by other article (Table S1. Anke Osterloh, MDPI Vaccines, 2022). No new tool is included in the table 3. Please revise the table with additional information such as provide any examples in vaccine development where particular tool is used or it can be used.
Author’s Response
We thank the reviewer for the comment and suggestion. Table 3 has been revised according to your valuable suggestion. We have now included detailed information about the new tools available for data mining and vaccine development. The details of the genomics method used for vaccine development and their status have also been added to table 4.
- Figure 1 looks adopted from Guangii Lu et al., 2020. Revise the diagram or provide reference to the legend of figure; ·Figure 2 is adopted from M.Lopez et al., 2015. Include more recent examples; Correct the sentences “pertussis toxins were developed”. Page 3· Correct sentences “Pathogen”. Page 3
Author’s Response
We thank the reviewer for their comments and suggestion. We have revised the diagram for figure 1 accordingly. For figure 2 we have included a reference. We also included more recent examples of RV with respect to intracellular bacterial pathogens. The sentences have also been corrected according to your valuable suggestions.
Again, we would like to thank the reviewer for their valuable feedback, suggestions, and support. We envisage the adequacy of our modifications. Thank you!
Round 2
Reviewer 1 Report
The revised manuscript from Khan et al is significantly improved. I would like to commend the amount of work the authors put into revising and adding content. I appreciate the grammatical corrections, as the manuscript now reads well. The changes to the figures were also a nice addition. There is significantly more discussion and explanations throughout. I have a few remaining minor corrections and comments.
Abstract: Italicize Helicobacter pylori. In the sentence “creation of effective vaccinations” change to “vaccines.”
Page 2, Paragraph 2: The parentheses with letters are not necessary. The abbreviation (S. aureus) are standard nomenclature. I do not understand this sentence: “This particular inconsistency in bacterial classification has since been resolved by researchers who examined the role of peptidoglycan recognition protein (PGRP) in Drosophila (Kurata,2010; Ferrand and Richard, 2013)”. How?
Page 3: Traditional vaccine development: Italicize species, and brucellosis is misspelled. Later on, “pathogen to loses its virulence…” should be “lose.”
Table 1: Check spacing between intravacuolar and pH. rickettsii should be italicized. M. leprae is also acid fast!
Table 2: C. burnetii (no d!).
Page 8, paragraph 1: ”Advancements in long reading sequencing technologies will facilitate the systematic assembly of diploid genomes, revolutionizing genomics by presenting the entire range of human genetic variation, covering some unaccounted heritability and uncovering novel disease processes”. I do not understand the point that is trying to be made with this sentence, or how it is relevant to vaccine development.
Make sure figures are the appropriate resolution.
Page 11-12: Viable (insert comma)but non-culturable cells (VBNC) cannot grow on the conventional medium (insert period). Therefore, novel approaches are…”
Page 12, paragraph 2: “However, such an immune response was not observed for serogroup B bacteria.”
Page 13: FACS stands for fluorescence-activated cell sorting.
Page 13 paragraph 2: (agalactiae, pyogenes, pneumoniae). Italicize and add “S.” to each (i.e. S. agalactiae). Same comment as above (page 2, paragraph 2, also in paragraph 3), remove the (H.). Some comments on the rewritten section:
The sharing of epitopes between host antigens and streptococcal bacteria is referred to as molecular mimicry….In this context, (add comma) the very first example was provided by Zabriskie in GAS (Zabriskie, 1967). The hallmark of rheumatic fever pathogenesis is molecular mimicry, where the GAS carbohydrate epitope, N-acetylglucosamine and streptococcal M protein structurally mimic cardiac myosin in human disease, leading to pancarditis and disease. In animal models immunized with the M protein of streptococcus and cardiac myosin (Guilherme et al., 2006). [This sentence seems like a fragment, was there something to discuss about these animal models?]. GAS vaccine candidate screening is based the conserved M epitope, N-terminal M peptides, cell surface and secreted proteins (Dale and Walker 2020)… vaccines are created….Vaccines based on the M protein (purified (spelled wrong) M protein…
Page 14: Paragraph 3: agalactiae should be lowercase.
Page 18, paragraph 2. “PPE19 is highly expressed in macrophages (per Mtb strain 2 copies)” Change to 2 copies/strain. Later: “The study carried out a pan genome investigation of Mtb to detrmine" Determine is spelled wrong.
Page 21, paragraph 2: “The main advantage of this approach is that it is a simple serum analysis method, however, a disadvantage is that not all proteins are expressed, resulting in impaired protein folding”. Should that be “resulting from impaired protein folding?” Meaning that the proteins are not expressed because they do not fold properly?
Page 23 paragraph 2: Define NGS (next generation sequencing) and SNPs (single nucleotide polymorphisms).
Page 23, paragraph 3: This statement: ” In the proteomics approach, 2D gel electrophoresis or mass spectrometry (MS) is used to reduce the complexity of samples by separating proteins using specific proteases.” is not entirely correct. Please rewrite as: For the proteomics approach, 2D gel electrophoresis or liquid chromatography (LC) are used to reduce the complexity of samples by separating proteins or peptides for analysis by mass spectrometry (MS).
Later in that paragraph, analyzed is spelled wrong. You should mention that the peak list is compared to a theoretical database of digested proteins identification.
Page 26: “The Actinobacterium A. mediterranei bacterium…” Should this say: “The bacterium Actinobacterium mediterranei”
Page 27, paragraph 1: I have never seen TDR. I think the proper term is pan drug resistant (PDR). These are bacteria that are resistant to every antibiotic.

Author Response
Reviewer 1
The revised manuscript from Khan et al is significantly improved. I would like to commend the amount of work the authors put into revising and adding content. I appreciate the grammatical corrections, as the manuscript now reads well. The changes to the figures were also a nice addition. There is significantly more discussion and explanations throughout. I have a few remaining minor corrections and comments.
Author’s Response
We thank the reviewer for the positive feedback. We will consider and implement the reviewers’ valuable suggestions for further improvement.
Abstract: Italicize Helicobacter pylori. In the sentence “creation of effective vaccinations” change to “vaccines.”
Author’s Response
We thank the reviewer for the suggestion. We have re-written the abstract to make it more concise and specific to the scope of our review.
Page 2, Paragraph 2: The parentheses with letters are not necessary. The abbreviation (S. aureus) are standard nomenclature. I do not understand this sentence: “This particular inconsistency in bacterial classification has since been resolved by researchers who examined the role of peptidoglycan recognition protein (PGRP) in Drosophila (Kurata,2010; Ferrand and Richard, 2013)”. How?
Author’s Response
We thank the reviewer for the comment. The parenthesis was removed, and the scientific names have now been stated according to standard nomenclature. The model organism Drosophila was used to resolve the inconsistencies within the bacterial classification system by examining the role of peptidoglycan recognition protein (PGRP) in triggering different immune response in different species of bacteria. Some minor details have been included in the revised manuscript.
Page 3: Traditional vaccine development: Italicize species, and brucellosis is misspelled. Later on, “pathogen to loses its virulence…” should be “lose.”
Author’s Response
We thank the reviewer for the comment. The required changes have been implemented.
Table 1: Check spacing between intravacuolar and pH. rickettsii should be italicized. M. leprae is also acid fast!
Author’s Response
We thank the reviewer for the comment. The required changes have been implemented.
Table 2: C. burnetii (no d!).
Author’s Response
We thank the reviewer for the comment. The mistake was rectified in the revised manuscript.
Page 8, paragraph 1:” Advancements in long reading sequencing technologies will facilitate the systematic assembly of diploid genomes, revolutionizing genomics by presenting the entire range of human genetic variation, covering some unaccounted heritability and uncovering novel disease processes”. I do not understand the point that is trying to be made with this sentence, or how it is relevant to vaccine development.
Author’s Response
We thank the reviewer for the comment. The development of sequencing methods has helped identify all human gene sequences. Previously, parts of the human genome that were difficult to examine have now been resolved using new technology, facilitating the assembly of diploid genomes. This will help identify disease processes in greater detail, further helping vaccine development.
Make sure figures are the appropriate resolution.
Author’s Response
We thank the reviewer for the comment. The resolution of the figures was corrected according to the journal requirements.
Page 11-12: Viable (insert comma) but non-culturable cells (VBNC) cannot grow on the conventional medium (insert period). Therefore, novel approaches are…”
Author’s Response
We thank the reviewer for the comment. As VBNC is the standard term, a comma has not been included. The period information of VBNC is included in the revised manuscript.
Page 12, paragraph 2: “However, such an immune response was not observed for serogroup B bacteria.”
Author’s Response
We thank the reviewer for the comment. The suggestion has been implemented.
Page 13: FACS stands for fluorescence-activated cell sorting.
Author’s Response
We thank the reviewer for the comment. The mistake was rectified in the revised manuscript.
Page 13 paragraph 2: (agalactiae, pyogenes, pneumoniae). Italicize and add “S.” to each (i.e. S. agalactiae). Same comment as above (page 2, paragraph 2, also in paragraph 3), remove the (H.).
Author’s Response
We thank the reviewer for the comment and suggestion. The scientific names have been Italicize and the genus name were has also been included in the revised manuscript.
Some comments on the rewritten section:
The sharing of epitopes between host antigens and streptococcal bacteria is referred to as molecular mimicry….In this context, (add comma) the very first example was provided by Zabriskie in GAS (Zabriskie, 1967). The hallmark of rheumatic fever pathogenesis is molecular mimicry, where the GAS carbohydrate epitope, N-acetylglucosamine and streptococcal M protein structurally mimic cardiac myosin in human disease, leading to pancarditis and disease. In animal models immunized with the M protein of streptococcus and cardiac myosin (Guilherme et al., 2006). [This sentence seems like a fragment, was there something to discuss about these animal models?]. GAS vaccine candidate screening is based the conserved M epitope, N-terminal M peptides, cell surface and secreted proteins (Dale and Walker 2020)… vaccines are created….Vaccines based on the M protein (purified (spelled wrong) M protein…
Author’s Response
We thank the reviewer for the suggestions. We have implemented the suggested changes where possible and have made additional modifications to help improve the flow of reading.
Page 14: Paragraph 3: agalactiae should be lowercase.
Author’s Response
We thank the reviewer for the highlighting this mistake. The mistake has been rectified in the revised manuscript.
Page 18, paragraph 2. “PPE19 is highly expressed in macrophages (per Mtb strain 2 copies)” Change to 2 copies/strain. Later: “The study carried out a pan genome investigation of Mtb to detrmine" Determine is spelled wrong.
Author’s Response
We thank the reviewer for the suggestions. The changes were incorporated in the revised manuscript and spelling mistakes were corrected.
Page 21, paragraph 2: “The main advantage of this approach is that it is a simple serum analysis method, however, a disadvantage is that not all proteins are expressed, resulting in impaired protein folding”. Should that be “resulting from impaired protein folding?” Meaning that the proteins are not expressed because they do not fold properly?
Author’s Response
We thank the reviewer for the suggestions. The sentence has been modified according to your valuable suggestions.
Page 23 paragraph 2: Define NGS (next generation sequencing) and SNPs (single nucleotide polymorphisms).
Author’s Response
We thank the reviewer for the suggestions. The suggested information has been included in the revised manuscript.
Page 23, paragraph 3: This statement: ” In the proteomics approach, 2D gel electrophoresis or mass spectrometry (MS) is used to reduce the complexity of samples by separating proteins using specific proteases.” is not entirely correct. Please rewrite as: For the proteomics approach, 2D gel electrophoresis or liquid chromatography (LC) are used to reduce the complexity of samples by separating proteins or peptides for analysis by mass spectrometry (MS).
Author’s Response
We thank the reviewer for the suggestions. The suggested information is included in the revised manuscript.
Later in that paragraph, analyzed is spelled wrong. You should mention that the peak list is compared to a theoretical database of digested proteins identification.
Author’s Response
We thank the reviewer for the suggestions. The information suggested is incorporated in the revised manuscript.
Page 26: “The Actinobacterium A. mediterranei bacterium…” Should this say: “The bacterium Actinobacterium mediterranei”
Author’s Response
We thank the reviewer for the suggestions and highlighting this mistake. The information was corrected according to your valuable suggestion.
Page 27, paragraph 1: I have never seen TDR. I think the proper term is pan drug resistant (PDR). These are bacteria that are resistant to every antibiotic.
Author’s Response
We thank the reviewer for the comment. With respect to TB, totally drug-resistant TB (TDR-TB) has been reported. We have now included a reference in the revised manuscript.
Reviewer 2 Report
The manuscript by Khan et al., has been improved, especially for inclusion of detailed information on history of traditional vaccine and genomics theory. Most of the major and minor remarks have also been considered. However, I have still few recommendations for further improvement. Here are key points to correct:
1. The key highlights and new addition to the previous published work is still missing in the entire article (especially abstract). Less effort has been observed to revise the abstract to synthesize new information. The abstract should be revised with concise and key information.
2. The minor errors pointed during the first revision is still not revised in many section of the manuscript, such as proper use of scientific terminology “ vaccine candidate” in the abstract line..” complete genomic database ..”.
3. The scientific nomenclature system for microorganism is revised but still lack correction. (Introduction: Escherichia (E.) coli etc).
4. Introduction section looks lengthy with lack of key concepts and required information. Key concept of article that is genomic approaches should keep in consideration and focus every aspect of the same while writing.
5. What do you mean by alternate classification of pathogen also exist? Inclusion of detailed information on traditional vaccine development is appreciated, but still lacks the key perceptive of genomic approaches. Section should be revised with any example of traditional vaccine development and how utilization of genomics approaches could improve the vaccine with any observed drawbacks.
6. Correct the sentence “The public databases for genome sequencing “genomes databases…”.
Author Response
Reviewer 2
The manuscript by Khan et al., has been improved, especially for inclusion of detailed information on history of traditional vaccine and genomics theory. Most of the major and minor remarks have also been considered. However, I have still few recommendations for further improvement. Here are key points to correct:
Author’s Response
We thank the reviewer for the positive feedback. We will consider and implement the reviewers’ valuable suggestions to further improve the manuscript.
- The key highlights and new addition to the previous published work is still missing in the entire article (especially abstract). Less effort has been observed to revise the abstract to synthesize new information. The abstract should be revised with concise and key information.
Author’s Response
We thank the reviewer for the positive feedback. We have re-written the abstract, to make it more concise and also to highlight the key issues and findings of the review.
- The minor errors pointed during the first revision is still not revised in many section of the manuscript, such as proper use of scientific terminology “ vaccine candidate” in the abstract line..” complete genomic database..”.
Author’s Response
Thank you for the suggestion. As stated in our response to point 1, we have re-written the abstract, being mindful of correct scientific terminology.
- The scientific nomenclature system for microorganism is revised but still lack correction. (Introduction: Escherichia (E.) coli etc).
Author’s Response
We thank the reviewer for the comment. We changed the nomenclature accordingly.
- Introduction section looks lengthy with lack of key concepts and required information. Key concept of article that is genomic approaches should keep in consideration and focus every aspect of the same while writing.
Author’s Response
We thank the reviewer for the comment. Yes, the section was extended during the revision in compliance with reviewer feedback requesting additional details. We have attempted to explain the basic concepts including bacterial classification, traditional methods for vaccine developments and the genomics revolution for vaccine development. We have also mentioned the details related to the role of genomics in vaccine development and the methods involved in different genomic approaches.
- What do you mean by alternate classification of pathogen also exist? Inclusion of detailed information on traditional vaccine development is appreciated, but still lacks the key perceptive of genomic approaches. Section should be revised with any example of traditional vaccine development and how utilization of genomics approaches could improve the vaccine with any observed drawbacks.
Author’s Response
We thank the reviewer for the comment and suggestion. There are other pathogen classification criteria in literature, apart from the classical method of classification. The classical methods primarily examine the infective nature of the pathogen but do not examine other characteristics of the pathogen. In the revised manuscript we have add some additional details. Additional information has also been added throughout the revised manuscript to help build key perceptive of genomic approaches in relation to vaccine development. Examples have also been included, as have details regarding the mechanisms of each genomic method.
- Correct the sentence “The public databases for genome sequencing “genomes databases…”.
Author’s Response
We thank the reviewer for the comment and suggestion. The sentence has been corrected according to your valuable suggestion.
Regards